# Inference-Aware Fine-Tuning for Best-of-N Sampling in Large Language Models

**Yinlam Chow**[†],[*] **Guy Tennenholtz**[‡], **Izzeddin Gur**[†], **Vincent Zhuang**[†], **Bo Dai**[†],

**Sridhar Thiagarajan**[†], **Craig Boutilier**[‡], **Rishabh Agarwal**[†], **Aviral Kumar**[†], **Aleksandra Faust**[†]

† Google Deepmind, ‡ Google Research

## Abstract

Recent studies have indicated that effectively utilizing inference-time compute is crucial for attaining better performance from large language models (LLMs). In this work, we propose a novel *inference-aware fine-tuning* paradigm, in which the model is fine-tuned in a manner that directly optimizes the performance of the inference-time strategy. We study this paradigm using the simple yet effective Best-of-N (BoN) inference strategy, in which a verifier selects the best out of a set of LLM-generated responses. We devise the first imitation learning and reinforcement learning (RL) methods for BoN-aware fine-tuning, overcoming the challenging, non-differentiable argmax operator within BoN. We empirically demonstrate that our BoN-aware models implicitly learn a meta-strategy that interleaves best responses with more diverse responses that might be better suited to a test-time input—a process reminiscent of the exploration-exploitation trade-off in RL. Our experiments demonstrate the effectiveness of BoN-aware fine-tuning in terms of improved performance and inference-time compute. In particular, we show that our methods improve the Bo32 performance of Gemma 2B on Hendrycks MATH from 26.8% to 30.8%, and pass@32 from 60.0% to 67.0%, as well as the pass@16 on HumanEval from 61.6% to 67.1%.

## 1 Introduction

An effective method for improving the performance of large language models (LLMs) is to leverage additional computation at inference-time: various works (Lightman et al., 2023; Wu et al., 2024; Kumar et al., 2024; Hosseini et al., 2024) have shown that by using search, re-ranking, multi-turn revision, and more generally, any approach that makes use of more tokens and inference-time compute, the performance of LLMs on various tasks can be significantly improved—so much that investing in improving inference-time computation might prove more beneficial than increasing model pre-training compute (Snell et al., 2024).

Despite this promise, existing work largely considers using inference-time computation as an optional post-hoc design choice, after conventional pre-training and fine-tuning. However, decoupling training and inference-time computation is *not* optimal; for example, if we knew that an LLM is allowed to make multiple attempts to solve a math problem, then it may be better to fine-tune it to explore diverse problem-solving strategies, rather than simply generating the candidates that represent the model's best attempt at solving the problem. Within the context of reasoning problems, these performance gains may be significant, as LLMs often fail due to their inability to draw complex inferences about the input and their internal knowledge (Chen et al., 2024).

We argue that the effectiveness of inference-time computation can be substantially increased by explicitly considering the inference procedure during training. We study this *inference-aware fine-tuning* paradigm using the Best-of-N (BoN) inference strategy, where the LLM generates multiple candidate responses, and a verifier selects the best one according to some scoring function (Cobbe et al., 2021). When this verifier is the ground-truth scoring function, BoN is equivalent to pass@N, a widely-used method for inference-time compute scaling (Brown et al., 2024). In contrast with traditional fine-tuning methods such as supervised fine-tuning (SFT) or reinforcement learning (RL),

---

*Correspondence to: `yinlamchow@google.com`

which are agnostic to the inference strategy used at test-time, our inference-aware (BoN-aware) methods directly optimize the performance of the BoN policy, and lead to significantly improved BoN performance at test time.

Our work makes several key contributions to the understanding and optimization of batch-of-neighbors (BoN) inference. (1) We formally define the inference-aware and BoN-aware problem setting, recognizing the crucial role of the inference strategy during training. (2) We establish a co-scaling behavior for BoN, quantifying the inherent trade-off between exploration and exploitation governed by the temperature ($T$) and the number of samples ($N$). This analysis directly informs the design and optimization of our BoN-aware algorithms. By understanding how these interactions influence performance, we can effectively tune these parameters to achieve an optimal balance between exploration and exploitation in both our supervised and reinforcement learning settings. (3) We develop a BoN-aware supervised fine-tuning algorithm that aligns the target distribution with the BoN policy distribution. (4) We extend our method to a general BoN-aware RL framework, enabling the policy to learn and solve downstream tasks under the BoN inference strategy. To further enhance BoN-aware fine-tuning, we devise specialized algorithms inspired by methods optimizing pass@$N$ accuracy, which promote implicit exploration and connect with established self-supervised learning techniques, particularly in scenarios where the environment reward can be used for verification. (5) Empirically, our experiments demonstrate the effectiveness of BoN-aware fine-tuning in terms of improved performance and inference-time compute. In particular, we show that our methods improve the Bo32 performance of Gemma 2B on Hendrycks MATH from 26.8% to 30.8%, and pass@32 from 60.0% to 67.0%, as well as the pass@16 on HumanEval from 61.6% to 67.1%.

## 2 INFERENCE-AWARE FINE-TUNING: A CASE STUDY WITH BEST-OF-N

Standard fine-tuning methods typically train LLMs to produce the best response for a given prompt. In LLM fine-tuning, a model (or *policy*) is trained via *supervised fine-tuning* (SFT), by maximizing the likelihood w.r.t. ground-truth data. Formally, we search for a policy $\pi : \mathcal{X} \mapsto \Delta_{\mathcal{Y}}$ that maximizes the likelihood $\mathbb{E}_{x \sim P, y \sim \pi^*(y|x)}[\log \pi(y|x)]$, where here, $\mathcal{X}$ and $\mathcal{Y}$ are the space of prompts and outputs of an LLM, $P$ is the prompt distribution, and $\pi^*$ is a distribution of expert responses. Alternatively, the policy can be fine-tuned via *reinforcement learning* (RL) (Schulman et al., 2017): $\max_{\pi \in \Pi} \mathbb{E}_{x \sim P, y \sim \pi(x)}[R(x, y)]$, to align the LLM's behaviors with the reward function $R(x, y)$. While popular, these methods have not taken the LLM's inference-time strategies into the account.

**Inference-Aware Fine-Tuning.** To address the gap between how LLMs are trained and how they are used at inference time, we develop inference-aware fine-tuning. During inference, the learned policy $\pi$ is often not directly used; rather some *inference strategy* $I : \Pi \times \mathcal{X} \mapsto \Delta_{\mathcal{Y}}$ is applied to it. For example, $I$ can be the BoN strategy, which samples multiple candidate responses, and selects the best using the score function of some verifier; or $I$ might be a search mechanism (Lightman et al., 2023) or self-correction (Kumar et al., 2024). To account for this inference strategy $I$, we alter the objective SFT and RL objectives to be "*aware*" of the inference strategy:

$$\max_{\pi \in \Pi} \mathbb{E}_{x \sim P, y \sim \pi^*(y|x)}[\log I(\pi, x)(y)], \text{ and} \qquad \text{(Inference-Aware SFT)}$$

$$\max_{\pi \in \Pi} J(\pi) := \mathbb{E}_{x \sim P, y \sim I(\pi, x)}[R(x, y)], \qquad \text{(Inference-Aware RL)}$$

Indeed, Inference-Aware SFT and Inference-Aware RL are aware of the strategy $I$. In what follows, we focus on the case where the inference strategy is BoN (i.e., $I \equiv \text{BoN}$), in both the SFT and RL setups. As we will later see, this brings about new algorithms for training the policy.

**BoN-Aware Problem Formulation.** We begin by defining the BoN strategy. This inference strategy samples $N$ resposnes from a model with some temperature $T$, and then selects the best one, based on some verifier score. Formally, the BoN inference policy can be written as:

$$I(\pi, x)(y) = \pi_{\text{bon}}(y|x; \pi, r, N, T) := \arg \max_{y' \in \{y_1, \dots, y_N\}} r(x, y'), \text{ s.t. } y_i \overset{T}{\sim} \pi(\cdot|x), x \in \mathcal{X}, \qquad (1)$$

where $\overset{T}{\sim}$ is a sample with temperature $T$, and $r : \mathcal{X} \times \mathcal{Y} \mapsto \mathbb{R}$ is a verifier score[1]. In what follows, when $r, N, T$ are clear from context, we write $\pi_{\text{bon}}(y|x; \pi)$. We see that the above strategy defines a class of BoN policies that is different from the learned policy $\pi$, demonstrating the gap

---

[1]The verifier score $r$ and the true reward $R$ can be related, or even equal, yet we do not make that assumption here. Usually, $r$ is a model trained to predict $R$, and therefore serves as a proxy of the true reward.

between training and inference. We inject this class of BoN polices into the Inference-Aware SFT and Inference-Aware RL frameworks to derive the instantiation of inference-aware fine-tuning.

Besides closing the gap between training and inference and mitigating potential discrepancies between the verifier score $r$ and the true reward $R$, BoN policies provide further benefits. The BoN mechanism introduces *implicit exploration* during training, bypassing the computational burden of explicit exploration (Cen et al., 2024). Selecting the best of $N$ samples allows the base policy to explore output variations, inducing a controlled exploration that can lead to more robust and generalizable policies, particularly in scaling behavior w.r.t. temperature $T$ and number of samples $N$.

Optimizing the BoN policy class is notoriously difficult due to the non-differentiability of the $\arg\max$ operator. Although several differentiable top-$k$ operators (Cuturi et al., 2019; Xie et al., 2020) might be exploited in $\pi_{\text{bon}}$, they induce approximation error, and more importantly, increase the computational cost dramatically. In our work, we derive a variational formulation of the learning problem w.r.t. $\pi_{\text{bon}}$ without top-$k$ operators, allowing us to construct novel algorithms for inference-aware BoN, using both standard supervised imitation learning (Section 3) and RL (Section 4).

**Exploration-Exploitation with BoN Sampling.** BoN sampling offers a natural mechanism for balancing exploration and exploitation in response generation. Adding BoN inference to the base model effectively explore the diverse possibilities within the model's output space while also exploiting its knowledge to generate high-quality candidates. This exploration-exploitation trade-off is crucial for solving various tasks and improving generalizability. To quantify such a trade-off, we empirically verify the implicit exploration and exploitation properties of BoN. We do this by revealing optimal co-scaling w.r.t. temperature $T$ and number of samples $N$. Specifically, for a fixed base policy $\pi$, at any prompt $x \in \mathcal{X}$ there is an optimal temperature $T^*(x)$ and optimal number of samples $N^*(x)$ which maximize performance of BoN:

$$N^*(x;\pi), T^*(x;\pi) \in \arg\max_{N,T} \mathbb{E}_{y \sim \pi_{\text{bon}}(y|x;\pi,r,N,T)}[R(x,y)].$$

To understand the connection between $N^*(x)$ and $T^*(x)$, we assess the performance of Gemma 2B (Team et al., 2024) on the MATH benchmark (Hendrycks et al., 2021), when applying the BoN inference strategy. Figure 1 shows empirical frequencies of problems, when varying $T^*(x)$ and $N^*(x)$ (larger marker size signifies higher frequency). The figure depicts a tradeoff between $T$ and $N$,

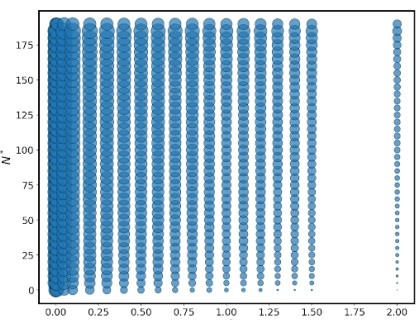

Figure 1: The relationship between the optimal number of samples ($N^*$) and optimal temperature ($T^*$) in BoN. The size of each marker at a given $(T, N)$ coordinate indicates the frequency of problems for which the $(T, N)$ pair resulted in the best BoN performance. The plot reveals a trade-off: "easier" problems have small $T^*$ and $N^*$, while "harder" problems require a larger $T^*$ for exploration and consequently a larger $N^*$.

reminiscent of the exploration-expoitation trade-off. When $T^*(x)$ is small, any $N$ is optimal (and particularly also a small $N$). These "easier" problems do not require heavy exploration (small $T^*$) and can therefore be more exploitative (small $N^*$). On the other hand, as $T^*$ increases, the base policy $\pi$ becomes more stochastic, resulting in more diversity and exploration. These more "difficult" problems, require more exploration (larger $T^*$), hence less exploitation (larger $N^*$). Indeed, in such cases, the distribution of $N^*$ shifts to high values. Our results suggest a tradeoff between exploration and exploitation, and further motivates the BoN-aware setup, which can account for this tradeoff uniformly across all samples.

Figure 1 also uncovers a cost-effective recipe for adjusting $T$ and $N$ for optimal BoN performance: we can learn how to fine-tune the model for better inference by simply adjusting these accessible parameters. However, it is important to note that relying solely on model selection has limitations. While this approach offers a computationally inexpensive way to improve BoN's inference-time performance, it may not fully capture the nuances of the LLM's behavior. With sufficient computational resources, general BoN-aware fine-tuning can further unlock performance gains by directly training the LLM to optimize for the exploration-exploitation trade-off of the BoN inference process.

## 3 SUPERVISED BON-AWARE FINE-TUNING

We begin by developing the BoN-aware SFT framework. Under this setting we assume we do not have access to the true reward, and only wish to maximize the likelihood of a dataset of expert

examples. Recall the definition of the BoN policy $\pi_{\text{bon}}$ in Equation (1). The Inference-Aware SFT version of BoN becomes:

$$\max_{\pi \in \Pi} \ \mathbb{E}_{(x,y) \sim \mathcal{D}} \big[ \log \pi_{\text{bon}}(y \mid x; \pi) \big], \tag{2}$$

A major difficulty in solving Equation (2) is the non-differentiability of the $\arg\max$ operator in the BoN procedure. To address this, we can use the variational approximation of $\pi_{\text{bon}}$ (see Section A.1)

$$\pi_{\text{bon}}(y|x) \propto \big[ \pi(y|x) \cdot \exp\left( \lambda_N Q_\pi(x, y) \right) \big], \tag{3}$$

where $Q_\pi(x, y) = \mathbb{E}_{y' \sim \pi(\cdot|x)} \big[ \mathbf{1}_{r(x,y) \geqslant r(x,y')} \big]$ is the expected *win-rate* over base policy $\pi$, characterizing the probability for which a response $y$ outperforms the responses generated by the base over the verifier score $r$. The constant $\lambda_N > 0$ is a solution of a 1D-search problem (Gui et al., 2024) (see details in Appendix A.1). It can be shown that $\lambda_N$ is monotonically increasing in $N$, and $\lambda_N \propto N$ approximately for large $N$. Plugging the variational form of Equation (3) into Equation (2) yields:

$$\max_{\pi \in \Pi} \mathbb{E}_{(x,y) \sim \mathcal{D}} \left[ \log \pi_{\text{bon}}(y|x) \right] := \mathbb{E}_{(x,y) \sim \mathcal{D}} \left[ \underbrace{\log \pi(y|x)}_{\text{Likelihood}} + \underbrace{\lambda_N \cdot Q_\pi(x, y)}_{\text{Inference-Awareness}} - \log Z_\pi(x) \right], \tag{4}$$

where $\quad Z_\pi(x) = \mathbb{E}_{\pi(y|x)} \left[ \exp\left( \lambda_N \cdot Q_\pi(x, y) \right) \right]$ is the partition function.

The above optimization problem reveals two term. While the first term tries to push the base policy $\pi$ into maximizing the likelihood of the data, the second term regularizes the policy to be more exploratory by increasing the data win rate over the policy. This in turn accounts for the sampling in BoN. For data efficiency when estimating the win rate $Q_\pi(x, y)$ we leverage a common practice in modeling pairwise preferences (Rafailov et al., 2023) to approximate the win rate with its "softened" counterpart: $Q_\pi(x, y) \approx \mathbb{E}_{y' \sim \pi(\cdot|x)} \left[ \sigma\left( r(x, y) - r(x, y') \right) \right]$, where $\sigma$ is the sigmoid function. Next, we exploit properties of policy gradient (Sutton et al., 1999) and the gradient of energy-based policies (Rafailov et al., 2024) to derive the gradient for Equation (4) (see Appendix A.2 for proof):

**Lemma 1 (BoN-SFT).** *The gradient of Equation* (4) *w.r.t. LLM parameters* $\theta \in \Theta$ *of* $\pi$ *is given by* $\mathbb{E}_{(x,y) \sim \mathcal{D}} \left[ \nabla_\theta f(x, y; \theta) \right] - \mathbb{E}_{x \sim \mathcal{D}, y \sim \pi_{bon}(\cdot|x)} \left[ \nabla_\theta f(x, y; \theta) \right]$, *where*

$$\nabla_\theta f(x, y; \theta) := \nabla_\theta \log \pi_\theta(y|x) - \lambda_N \cdot \mathbb{E}_{y' \sim \pi_\theta} \left[ \nabla_\theta \log \pi_\theta(y'|x) \cdot \mathbf{1}\{r(x, y) < r(x, y')\} \right]. \tag{5}$$

Our formulation circumvents the non-differentiability of the BoN distribution, allowing solution of BoN-SFT via standard gradient-ascent algorithms. The individual terms of the gradient imply the following: **(1)** $\pi$ clones the expert behavior by maximizing its likelihood over $\mathcal{D}$; **(2)** it aligns with the verifier score ranking, which assigns a high win-rate to the expert over the base; **(3)** it avoids over-fitting by limiting its likelihood over the BoN sample; and **(4)** it maintains overall response quality by reducing the win rate between its best and average samples.

## 4 BoN-Aware Fine-Tuning Using Reinforcement Learning

Training LLMs that are amenable to BoN sampling can be framed within the RL paradigm, which trains an agent (LLM) that optimizes its actions within an environment. In this context, the LLM generates N responses (candidates actions) for a given prompt (contexts). A separate macro agent (verifier) selects the candidate deemed most suitable according to a predefined criterion (e.g., probability of success). This action is then deployed to the environment, yielding a reward (e.g., task completion). The key challenge in training this agent lies in achieving two objectives simultaneously: (i) Enhancing agent's exploration capabilities to generate diverse candidates that cover the space of potential solutions and align with the verifier's preferences; (ii) Maximizing the environment reward of the final response. Motivated by this observation, we utilize RL for BoN-aware fine-tuning, enabling the development of more robust and adaptable LLMs. A schematic of the BoN-Aware RL framework is shown in Figure 2.

The BoN-Aware RL problem takes the following form:

$$\max_{\pi \in \Pi} J(\pi) := \mathbb{E}_{x \sim P, y \sim \pi_{\text{bon}}(\cdot|x; \pi, r, N, T)}[R(x, y)]. \tag{6}$$

We train the BoN policy $\pi_{\text{bon}}$ (paramterized by $\pi$) to attain a high environment reward. Apart from enabling better exploration, using the environment reward $R(x, y)$ in BoN-RL allows the base policy to tolerate potential errors in the verifier $r(x, y)$. We first develop a general algorithm for solving the BoN-aware RL problem. We then study an important subclass which assumes a binary reward, a common feature of many reasoning problems (e.g., math, code).

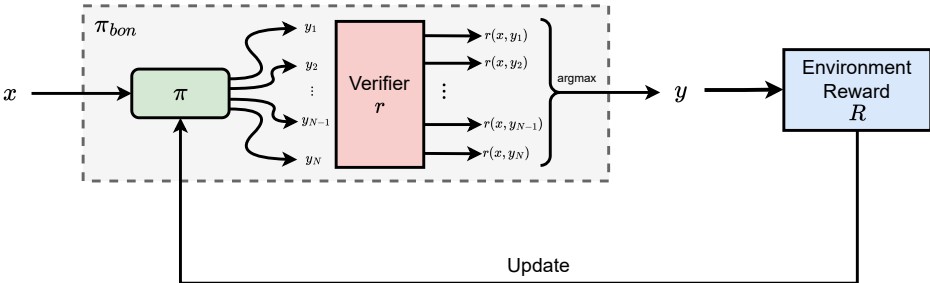

Figure 2: BoN-Aware RL fine-tuning, where $N$ independent samples are first drawn from base $\pi$ and ranked by the verifier $r$. The BoN policy then optimizes the environment reward of the BoN samples (Lemma 2).

We begin with deriving a gradient estimator to the objective in Equation (6). Exploiting the connection between the BoN policy and its energy-based policy counterpart in Equation (10), and using derivations analogous to those in Lemma 1, we compute the gradient of $J(\theta)$, which leads to a REINFORCE-style algorithm (Williams, 1992) (see Appendix A.3 for proof):

**Lemma 2 (BoN-RL).** *The gradient of Equation (6) w.r.t. LLM parameters $\theta \in \Theta$ of $\pi$ is given by*

$$\nabla_\theta J(\theta) = \mathbb{E}_{x\sim\mathcal{D}, y\sim\pi_{bon}(\cdot|x)} \left[ \nabla_\theta \log \pi_\theta(x, y) \cdot (R(x, y) - b(x)) \right]$$
$$- \lambda_N \cdot \mathbb{E}_{x\sim\mathcal{D}, y\sim\pi_{bon}(\cdot|x), y'\sim\pi_\theta} \left[ \nabla_\theta \log \pi_\theta(y'|x) \cdot \mathbf{1}\{r(x, y) < r(x, y')\} \cdot (R(x, y) - b(x)) \right], \quad (7)$$

*where $b(x) = \mathbb{E}_{y\sim\pi_{bon}(\cdot|x)}[R(x, y)]$ is a baseline for variance reduction (Schulman et al., 2015).*

This formulation resembles the standard REINFORCE gradient with the main difference of drawing samples from the BoN policy (instead from the base policy $\pi$). This allows one to solve BoN-RL via actor-critic methods (Sutton et al., 2009a). In practice, one can replace $b(x)$ with a learned value baseline $b_\psi(x)$ parameterized by $\psi$, for which $\psi$ is updated by gradient descent w.r.t. the critic value loss. While BoN-RL inherits the benefits of verifier alignment from BoN-SFT, and can be viewed as a reward-weighted variant of the popular STaR method (Zelikman et al., 2022), generally it can be rather sample inefficient (especially when $N$ is large), as estimating both the value function $b(x)$ and the policy gradient in BoN-RL require samples from the BoN distribution. See Appendix C for a discussion on alleviation using BoN distillation (Sessa et al., 2024).

**BoN-RL with Binary Reward and Verifier.** While Lemma 2 provides a general method for BoN-aware RL, the policy gradient estimator in Equation (7) is sample inefficient for a general rewards and verifiers. However, many domains admit binary success/failure metrics (e.g., reasoning tasks, math, coding) which allow an efficient gradient estimator, obviating the need for value estimation. Specifically, with a binary reward known to the verifier, i.e., $R(x, y) = r(x, y) \in \{0, 1\}$, Theorem 1 of Sessa et al. (2024) implies the following closed-form solution of the BoN policy $\pi_{\text{bon}}$:

$$\pi_{\text{bon}}(y|x) = \begin{cases} \pi(y|x) \cdot P_{\text{fail}}(x)^{N-1} & \text{if } R(x, y) = 0 \\ \frac{\pi(y|x)}{1 - P_{\text{fail}}(x)} \cdot \left(1 - P_{\text{fail}}(x)^N\right) & \text{if } R(x, y) = 1 \end{cases}, \quad (8)$$

where $P_{\text{fail}}(x) := \mathbb{E}_{y'\sim\pi(\cdot|x)} \left[ \mathbf{1}_{R(x,y')=0} \right]$ is the fraction of problems on which the base policy $\pi$ is incorrect. Under the binary assumption, $\pi_{\text{bon}}$ is a weighted distribution of the base policy $\pi$, whose importance sampling ratio depends on the its failure probability $P_{\text{fail}}(x)$. Introducing this closed form of $\pi_{\text{bon}}$ to Lemma 2, we obtain the following policy gradient (see Appendix A.4 for proof):

**Lemma 3 (BoN-RLB).** *Assume $R(x, y) \in \{0, 1\}$. The gradient of Equation (6) w.r.t. LLM parameters $\theta \in \Theta$ of $\pi$ is given by*

$$\mathbb{E}_{x\sim\mathcal{D}} \left[ \mathbb{E}_{y\sim\pi_{bon}, R=1} \left[ \nabla_\theta \log \pi_\theta(y|x) \right] \cdot g_N^+(P_{fail}(x)) + \mathbb{E}_{y\sim\pi_{bon}, R=0} \left[ \nabla_\theta \log \pi_\theta(y|x) \right] \cdot g^-(P_{fail}(x)) \right],$$

*where the positive and negative sample-dependent weights are given by*

$$g_N^+(p) = \frac{N \cdot p^{N-1}}{1 - p^N}, \quad g^-(p) = \frac{N \cdot p}{1 - p}. \quad (9)$$

Lemma 3 not only reveals an efficient policy gradient estimator for binary reward, but more importantly demonstrates how BoN-RLB balances positive and negative examples in its gradient update. It proposes novel way to re-weigh BoN-RLB's training examples, which prioritizes harder examples (as $P_{\text{fail}}(x) \to 1$) by giving their positive samples exponentially more influence and aggressively redistributing log likelihood away from incorrect responses. The significance of this asymmetric

Table 1: Summary of Our Best-of-N-Aware Fine-Tuning Methods

| Method | Offline Data | Reward (R) | Verifier (R≠r) | Binary (R=r) | Positive Only | Closed Form |
|--------|:---:|:---:|:---:|:---:|:---:|:---:|
| BoN-SFT | ✓ | ✗ | ✓ | ✗ | ✓ | ✗ |
| BoN-RL-V | ✗ | ✓ | ✓ | ✓ | ✗ | ✗ |
| BoN-RL-S | ✗ | ✓ | ✗ | ✓ | ✗ | ✗ |
| BoN-RLB | ✗ | ✓ | ✗ | ✓ | ✗ | ✓ |
| BoN-RLB(P) | ✗ | ✓ | ✗ | ✓ | ✓ | ✓ |

weighting scheme is that it infuses *implicit exploration* capabilities to the base policy. As Tajwar et al. (2024) observed, when the model reduces the likelihood of negative responses, it shifts that probability mass towards a mode of the learned policy – essentially reinforcing existing successful strategies (exploitation). However, if these high-likelihood regions later produce errors, the resulting negative gradient redistributes this mass again, pushing the model to explore other potential solutions. This iterative process of concentrating probability mass and subsequent redistribution through negative gradients drives a form of exploration: as long as the model can produce correct solutions, it need not devote most of its sampling budget $N$ on that but can also explore more diverse solutions.

**Positive-only Weighting.** Although we have illustrated the benefits of an asymmetric weighting scheme in BoN-RLB for exploration, training with both positive and negative examples may be infeasible (e.g., in a data-limited online RL system that only records positive examples). To tackle this, we apply a change of measure to Lemma 3 with the BoN distribution to derive a policy gradient that only involves positive examples (see Appendix A.5 for proof):

**Corollary 4 (BoN-RLB(P)).** *Assume $R(x, y) \in \{0, 1\}$. The gradient of Equation (6) w.r.t. LLM parameters $\theta \in \Theta$ of $\pi$ is given by $\mathbb{E}_{x \sim \mathcal{D}}[\mathbb{E}_{y \sim \pi_{bon}, R=1}[\nabla_\theta \log \pi_\theta(y|x)] \cdot \overline{g}_N^+(P_{fail}(x))]$, where $\overline{g}_N^+(p) := \frac{N \cdot p^{N-1} \cdot (1-p)}{(1-p^N)}$.*

Notice that the weighting $\overline{g}^+(p, N)$ is monotonically increasing in $p \in [0, 1]$ and lies within $[0, 1]$ for any $N$. Using this gradient update, BoN-RLB(P) resembles a weighted version of BoN-STaR (see Remark C.3 in the appendix), where it clones positive examples generated by the current BoN policy and up-weights the more difficult ones, where $P_{fail}(x)$ is close to 1.

Table 1 summarizes different BoN-aware fine-tuning methods including: (1) **BoN-SFT** from Lemma 1, (2) BoN-RL from Lemma 2 with a verifier, **BoN-RL-V**, (3) BoN-RL with environment reward as verifier, **BoN-RL-S**, (4) **BoN-RLB** from Lemma 3, and (5) **BoN-RLB(P)** from Corollary 4. These methods are designed to leverage the BoN selection strategy during model fine-tuning, under different settings mentioned in Sections 3 and 4. BoN-SFT uses expert data but doesn't involve explicit reward learning. It relies on a verifier to select the best output and only trains with positive examples. BoN-RL-V and BoN-RL-S employ RL to optimize BoN performance, with the former using a verifier for response selection while the latter relying solely on a reward signal. BoN-RLB and BoN-RLB(P) also utilize RL on the special case of binary reward feedback ($R = r \in \{0, 1\}$). They both have closed-form solutions, which can lead to more efficient learning. BoN-RLB(P) only learns from positive examples, which might be beneficial for (e.g., safety-critical) situations where collecting negative data is difficult.

## 5 EXPERIMENTS

In this section we address the following questions: (1) Can we quantify the co-scaling relationship between the BoN number of samples $N$ and temperature $T$, enabling joint optimization of these parameters? (2) Do inference-aware fine-tuning methods (SFT and RL) enhance the effectiveness of BoN sampling? (3) Do these improvements generalize across problem domains and BoN settings?

### 5.1 CO-SCALING ANALYSIS OF SAMPLE SIZE $N$ AND TEMPERATURE $T$ IN BON

We study the co-scaling behaviors of BoN and pass@$N$ performance over varying $N$ and $T$ by experimenting pre-trained Gemma 2B and 9B models on the Hendrycks MATH benchmark (Hendrycks et al., 2021). Our experiments showed consistent co-scaling behaviors across different model sizes, therefore we summarize the results of 2B model below and include the 9B findings in Appendix D.1. As illustrated in Figure 3 (Figure 9, Figure 10 in Appendix D.1), pass@$N$ consistently increases with higher $N$, as commonly observed (Brown et al., 2024). Our analysis suggests that this relationship can be captured by a power-law function of the following form:

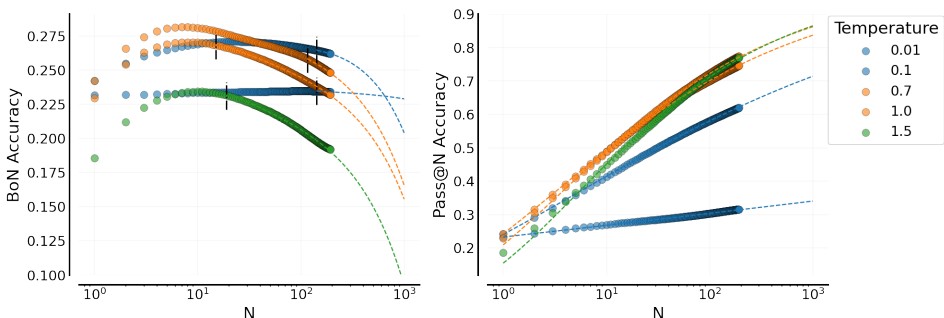

Figure 3: BoN and pass@$N$ performance of Gemma 2B policy and reward models w.r.t. varying $N$ and $T$. pass@$N$ monotonically improves with $N$; BoN shows inflection points as $N$ increases. Colored dashed lines denote predictions of scaling functions; black dashed lines in BoN plot denote the last inflection points.

$pass@N(T) \approx \exp(a(T)N^{b(T)})$, where the parameters $a(T)$ and $b(T)$ are temperature-dependent and derived by fitting the model to data at a specific temperature $T$. Further analysis of this scaling behavior (detailed in Appendix D.1) indicates that there is a strong positive correlation between the optimal temperature and $N$, which aligns with the intuition that larger $N$ benefits from broader exploration (higher $T$), while smaller $N$ favors focused exploitation (constant $T$). This relationship is straightforward, as there are no verifier errors confounding the selection of best responses.

Our experiments demonstrate an intriguing relationship between BoN accuracy, $T$, and $N$. We find that lower temperatures generally yield better BoN accuracy. Furthermore, BoN accuracy generally decreases as $N$ increases, but degrades more rapidly with higher temperatures. With larger $N$ and $T$, the increased randomness in the base policy inherently generates more "bad" samples (with poor accuracy). This phenomenon suggests that the verifier is sensitive to noise and may mistakenly select random outputs generated at higher temperatures as the best responses due to misalignment with the true reward (Type II error). Conversely, at very low temperatures, BoN accuracy improves with $N$, indicating the algorithm remains in an exploitation phase. Optimal performance is observed at moderate $N$ values, striking a balance between exploration and exploitation.

## 5.2 INFERENCE-AWARE FINE-TUNING WITH BON SAMPLING

**Experimental setup.** We study 2B and 9B Gemma 2 models (Team et al., 2024) on canonical mathematical problem-solving and code generation benchmarks. For math, we train and evaluate on Hendrycks MATH benchmark (Hendrycks et al., 2021), and additionally evaluate on two held-out math benchmarks: Functional MATH (Srivastava et al., 2024) and MathOdyssey (Fang et al., 2024). For coding, we train on MBPP (Austin et al., 2021), and evaluate on HumanEval (Chen et al., 2021). More details of our experiments can be found in Appendix D.2.

Our main evaluation metrics are **BoN** and **pass@N** accuracies, i.e. the accuracies of the policies defined in Equation (1) with a learned verifier and ground-truth reward respectively. For math, we consider both metrics, using a learned verifier for BoN, and for coding, we consider only pass@$N$ because test-case feedback is usually available.

We benchmark our proposed methods BoN-SFT, BoN-RL-V, BoN-RL-S, BoN-RLB, and BoN-RLB(P) against several baselines: (1) STaR from Remark C.3, which uses self-training over correctly generated responses; (2) RL (Lee et al., 2023) with verifier feedback (RL-V); (3) RL with environment feedback (RL-S); (4) standard SFT of the base policy (SFT, $N' = 1$); and (5) BoN with the pre-trained model (Base model). We denote by $(N', T')$ and $(N, T)$ the number of samples and temperature used in training and evaluation, respectively. We use $T = T' = 1.0$ unless specified otherwise. Similar to co-scaling, our experiments show similar trends with 2B and 9B models, therefore we summarize that of the 2B model below and defer the 9B results to Appendix D.3.

**BoN-aware supervised fine-tuning.** We first evaluate the BoN and pass@$N$ performance of offline SFT methods, including BoN-SFT with various $N'$, and base SFT ($N' = 1$), with results shown in Figure 4. We find that base SFT significantly degrades upon the base model, indicating that it causes overfitting or lack of generalization. BoN-SFT is able to improve the BoN accuracy significantly, especially with increasing $N'$. We find that BoN-SFT with $N' = 32$ achieves the best performance for both BoN accuracy and pass@$N$, suggesting that it is able to produce both high-

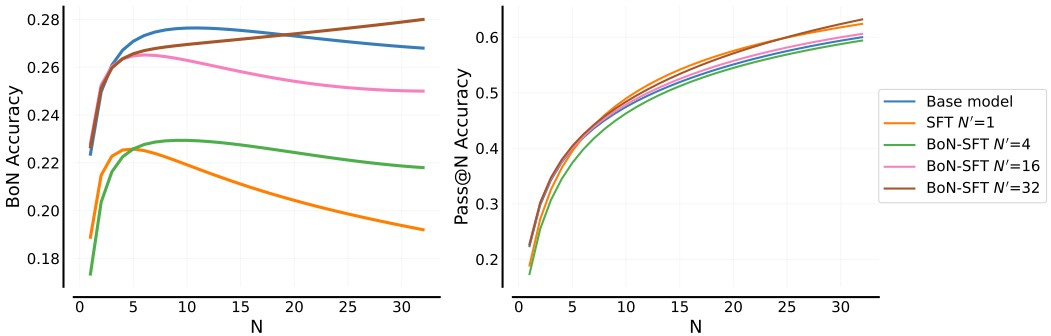

Figure 4: BoN Accuracy and pass@$N$ accuracy for BoN-SFT with Gemma 2B models.

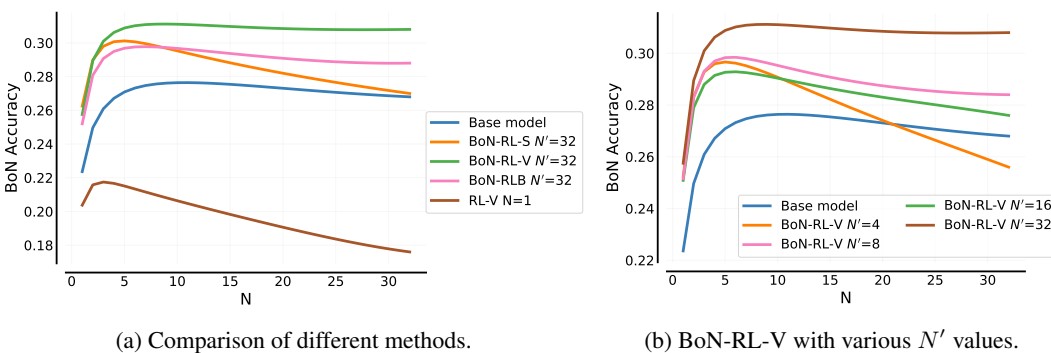

(a) Comparison of different methods.

(b) BoN-RL-V with various $N'$ values.

Figure 5: BoN accuracy on BoN-RL methods and baselines, with Gemma 2B model on MATH.

quality and exploratory responses. To improve substantively over the base model, we next turn to RL, which should be more effective by virtue of being on-policy.

**BoN-RL-V improves BoN Accuracy.** Our BoN RL algorithm, BoN-RL-V, with $N' = 32$, significantly outperforms several baselines (Figure 5a). Specifically, it boosts the Bo-32 accuracy of our base model from $26.8\%$ to $30.8\%$. As expected, the inference-unaware RL-V method performs poorly, likely due to common reward hacking issues (Jinnai et al., 2024). We also find that training with the same verifier used as test-time is critical - our other proposed inference-aware methods that use the environment reward instead of the verifier (BoN-RL-S and BoN-RLB) show improvement over the base model but are substantially worse than BoN-RL-V.

We plot the performance of BoN-RL-V with different training $N'$ in Figure 5b, and observed the best performance when training with 32 samples ($N' = 32$). Interestingly, although the gains are more pronounced at higher $N$ values, training with large $N'$ leads to consistent improvements across all $N = 1$ to 32. This suggests that RL-BoN-V not only optimizes the specific BoN setting it was trained on, but also generalizes to other BoN configurations, including large improvements on direct $N = 1$ accuracy (from $22\%$ to $26\%$). We hypothesize that the RL-BoN-V method significantly enhances BoN accuracy by effectively exploring a larger sample space during training. Training with a large $N'$ allows the base policy to explore a wider range of responses to generate higher-quality responses, similar to how effective exploration in RL leads to better performance and generalization across different scenarios.

**BoN-RL-S, BoN-RLB, and BoN-RLB(P) improve pass@$N$.** We next analyze the pass@$N$ performance of various methods on math and coding benchmarks, aiming to understand how different training approaches impact the models' ability to generate diverse and correct solutions. As shown in Figure 6, our inference-aware methods designed to explicitly optimize pass@$N$ during training (BoN-RL-S, BoN-RLB, or BoN-RLB(P)) can lead to better test-time pass@N across both domains, highlighting the importance of directly considering the desired evaluation metric during training. Notably, in the MATH domain (Figure 6a), BoN-RL-S significantly improves the pass@32 of Gemma 2B from $60.0\%$ to $67.0\%$. This suggests that by encouraging the model to explore a wider range of solution strategies during training, we can substantially enhance its ability to generate correct answers. Conversely, standard RL with $N' = 1$ slightly degrades pass@32, indicating

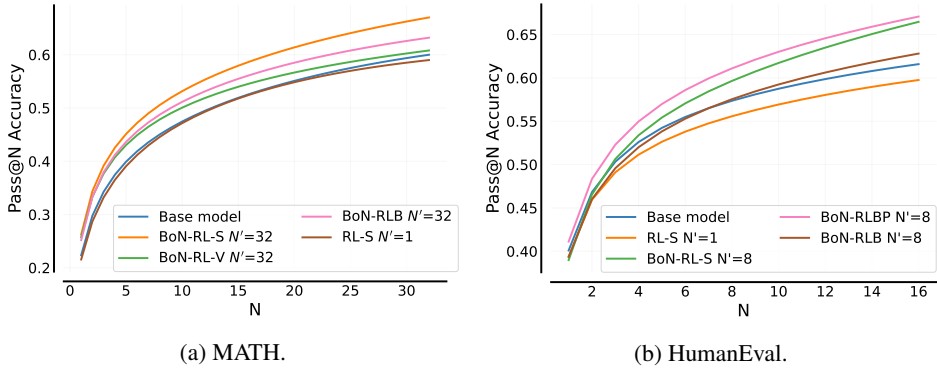

(a) MATH.    (b) HumanEval.

Figure 6: pass@$N$ of BoN-RL methods with binary reward as verifier, with Gemma 2B models.

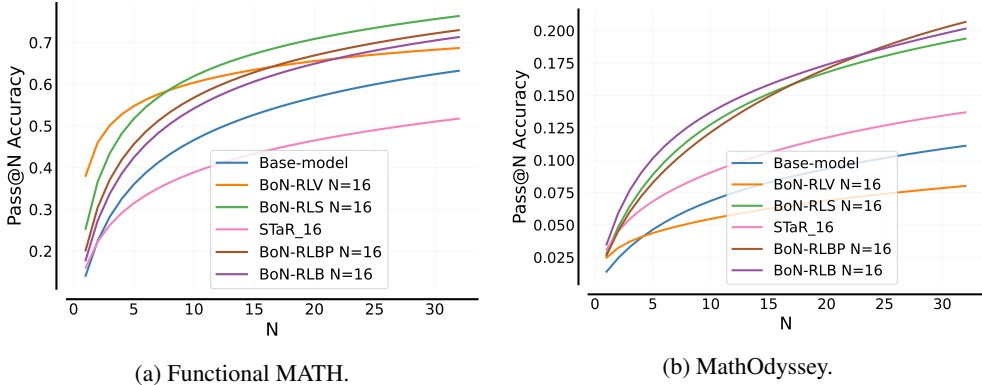

(a) Functional MATH.    (b) MathOdyssey.

Figure 7: pass@$N$ of BoN-RL and baselines on held-out benchmarks with Gemma 2B models.

that over-optimizing for immediate performance can actually hinder the model's exploration capabilities and limit its overall problem-solving ability. Interestingly, BoN-RL-V does not significantly improve pass@$N$, likely because its training objective focuses on robustness to a verifier, which may not perfectly align with generating diverse correct solutions. Similar trends are observed on coding benchmarks (Figure 6b), where pass@$N$-aware methods, particularly BoN-RLBP, significantly outperform the base model, increasing the pass@16 performance of Gemma 2B from $61.6\%$ to $67.1\%$. Again, standard RL fine-tuning ($N' = 1$) negatively impacts performance, reducing pass@16 to $59.8\%$, demonstrating the downsides of solely focusing on reward maximization during training

**Generalization of BoN-RL to held-out benchmarks and other temperatures.** To further assess the generalization capabilities of our BoN aware fine-tuned models, we evaluate their performance on two challenging, held-out benchmarks: Functional Math (Srivastava et al., 2024) and MathOdyssey (Fang et al., 2024). As shown in Figure 7, our fine-tuned models demonstrate clear improvements on both benchmarks. Our BoN-aware policies also generalize well to different sampling settings. As shown in Figure 8, BoN-RL-V (trained with $T' = 1.0$) consistently outperform the base model across all evaluation temperatures ($T \in \{0.1, 1.0, 1.5\}$). Similarly, BoN-RL-S show superior performance for pass@$N$ at all temperatures, with increasing gains at higher temperatures. This highlights the benefits of broader exploration, even after BoN-aware training. For BoN-accuracy, lower temperatures favored both models, but BoN-RL-V consistently excelled, particularly at lower temperatures, demonstrating its generalizability across different exploration-exploitation regimes. Furthermore, BoN-RL-V show greater resilience to accuracy degradation at higher temperatures, suggesting an enhanced ability to adapt to verifier failure modes and mitigate Type-II errors.

## 6    RELATED WORK

Large language models (LLMs) can leverage inference-time computation to improve the quality of their generated outputs (Welleck et al., 2024), particularly on reasoning tasks. One common approach is to use chain-of-thought (Wei et al., 2022), where the model generates a step-by-step rationale before generating the final output. Another useful approach that can be combined with

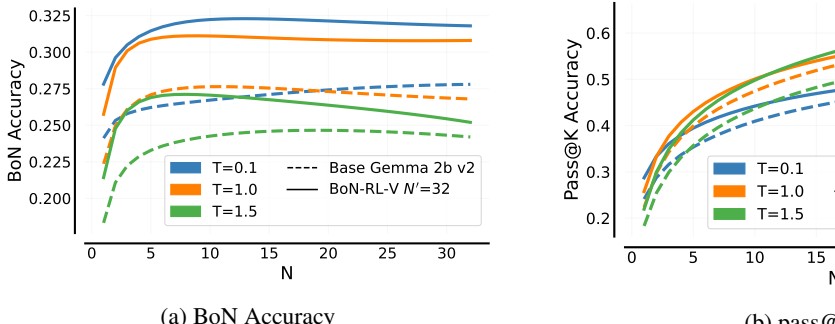

(a) BoN Accuracy

(b) pass@$N$

Figure 8: BoN and pass@$N$ over various temperatures. BoN-RL with verifier and exact reward (solid lines) are trained with $T = 1.0$. Dashed and solid lines show the base (Gemma 2B) and finetuned models respectively.

chain-of-thought is Best-of-N rejection sampling (Charniak & Johnson, 2005; Stiennon et al., 2020), which is our focus in this work. In Best-of-N, we generate multiple candidate outputs from an LLM and select the best output. BoN re-ranking can be done either using oracle information, such as checking final answers for solving math problems, which is also known as pass@$N$ (Chen et al., 2021), or learned verifiers (Cobbe et al., 2021; Lightman et al., 2023; Hosseini et al., 2024; Zhang et al., 2024). Recent work also empirically analyzes strategies that optimally trade off additional test-time compute for improved performance (Wu et al., 2024; Snell et al., 2024).

Closely related to our approach is prior work that fine-tunes LLMs to improve their self-correction capabilities (Kumar et al., 2024; Snell et al., 2024) or search capabilities on planning tasks (Gandhi et al., 2024; Lehnert et al., 2024), which allows for more efficient scaling with test-time compute. By contrast, our work focuses on inference-aware fine-tuning that directly optimizes for Best-of-N performance, instead of an intermediate capability that be used at test-time.

To make an LLM amenable to test-time scaling, techniques like STaR (Zelikman et al., 2022) or ReST$^{EM}$ (Singh et al., 2023) have been employed to fine-tune the model using on-policy data. This process leverages BoN sampling to iteratively generate better responses, and fine-tunes on this curated data, for which the LLM learns to improve its proposal distribution, effectively increasing the likelihood of generating high-quality outputs during inference.

Finally, our work is related to recent work on leveraging tree search to enhance decision-making in RL (Dalal et al., 2021). A key challenge in both BoN sampling and tree search lies in mitigating the impact of imperfect value estimation. Dalal et al. (2021) address this in tree search by penalizing actions leading to states with high $Q$-value error, making inference more pessimistic for out-of-distribution samples. In contrast, we address BoN verifier error during training, learning responses robust to these errors and aligning training with inference. Our framework extends to tree search, using the verifier as an approximate $Q$-function and optimizing policy robustness to its errors.

## 7 CONCLUSION

We introduced inference-aware fine-tuning, a novel paradigm that bridges the gap between training and inference for LLMs. Specifically for the Best-of-N inference strategy, we discovered a co-scaling law for BoN that guides the optimization of temperature and sample size, developed a gamut of fine-tuning algorithms that handle various imitation learning and reinforcement learning settings, training LLMs to generate diverse and high-quality outputs tailored for BoN inference, demonstrated the efficacy of these methods by significantly improving on BoN accuracy and pass@$N$ on the standard MATH reasoning benchmark over state-of-the-art baselines, highlighting the robustness and generalizability of our approaches across various BoN configurations.

Our work exemplified how BoN-aware fine-tuning learns a meta-strategy, which interleaves best responses with more diverse responses that might be better suited for BoN sampling. These findings underscore the potential of inference-aware fine-tuning to unlock previously undiscovered capabilities in LLMs through aligning training methodologies with inference-time compute strategies. Future work includes extending this framework to incorporate more complex, inference algorithms (e.g., reasoning, critique-and-revise, MCTS), developing contextual BoN-aware algorithms that can generalize to various tasks, investigating the interplay between the co-scaling of temperature, sample size, and BoN-aware fine-tuning, and applying our algorithms to more larger-scale problems.

**Reproducibility Statement.** We utilize the publicly available Gemma 2B and 9B language models, the Hendrycks MATH benchmark, and the HumanEval coding benchmark – all accessible to the research community. Our experimental setup is described in detail in Section 5. Furthermore, the appendix provides comprehensive pseudo-code (Algorithms 1 to 4) and implementation details for our BoN-aware fine-tuning algorithms (BoN-SFT, BoN-RL, BoN-RLB, and BoN-RLB(P)). We also delve into the theoretical underpinnings of our methods in the main text and the appendix, enabling a thorough understanding of our approach.

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

# A THEORETICAL DERIVATIONS

## A.1 VARIATIONAL APPROXIMATION OF BON

We assume that the verifier score $r(x, y)$ is unique for all $x, y$, and the base model $\pi$ has a finite set of possible outcomes for each context (Beirami et al., 2024).

**Proposition 5** (Theorem 2 in Gui et al. (2024)). *With negligible error, one may effectively approximate $\pi_{bon}$ as the solution to the following optimization problem:*

$$\pi_{bon}(y|x) \in \arg\max_{\mu(\cdot|x) \in \Delta_{\mathcal{Y}}} \mathbb{E}_{y \sim \mu} [Q_\pi(x, y)] - \frac{1}{\lambda_N} KL(\mu||\pi)(x), \tag{10}$$

*where $Q_\pi(x, y) = \mathbb{E}_{y' \sim \pi(\cdot|x)} \left[ \mathbf{1}_{r(x,y) \geqslant r(x,y')} \right]$ is the expected* win-rate *over $\pi$, and*

$$\frac{(\lambda_N - 1) \exp(\lambda_N + 1)}{\exp(\lambda_N - 1)} - \log\left(\frac{\exp \lambda_N - 1}{\lambda_N}\right) = \log N - \frac{N - 1}{N}, \tag{11}$$

*through $\lambda_N$ scaling sub-linearly with BoN number of samples $N$.*

We can show the optimal solution to Equation (10) has a closed form $\pi^*_{bon} \propto [\pi \cdot \exp(\lambda_N Q_\pi)](y|x)$. This can also be revealed by viewing Equation (10) as the variational form of Bayes' rule (Williams, 1980; Zellner, 1988; Zhu et al., 2014; Dai et al., 2016), whose optimal solution is the posterior. This implies $\pi_{bon}$ can be represented by an exponential-twisting policy (Gerber et al., 1993) over base policy $\pi$ with energy function $\lambda_N \cdot Q_\pi(y, x)$, partition function $Z_\pi(x) = \mathbb{E}_{\pi(y|x)} [\exp(\lambda_N \cdot Q_\pi(x, y))]$, and an appropriate $\lambda_N$ from Equation (11).

In this section we will provide proofs for the technical results in this paper.

## A.2 PROOF OF LEMMA 1

Therorem 2 of Gui et al. (2024) shows that, with negligible error, one may effectively approximate $\pi_{bon}$ as the solution to the following optimization problem:

$$\pi_{bon}(y|x) \in \arg\max_{\mu(\cdot|x) \in \Delta_{\mathcal{Y}}} \mathbb{E}_{y \sim \mu} [Q_\pi(x, y)] - \frac{1}{\lambda_N} KL(\mu||\pi)(x), \tag{12}$$

where $Q_\pi(x, y) = \mathbb{E}_{y' \sim \pi(\cdot|x)} \left[ \mathbf{1}_{r(x,y) \geqslant r(x,y')} \right]$ is the expected *win-rate* over $\pi$, and this yields the variational form $\pi_{bon} \propto [\pi \cdot \exp(\lambda_N Q_\pi)](y|x)$. Plugging the variational form of $\pi_{bon}$ into (2) yields the learning problem for $\pi$:

$$\max_{\pi \in \Pi} \mathbb{E}_{(x,y) \sim \mathcal{D}} [\log \pi_{bon}(y|x)] := \mathbb{E}_{(x,y) \sim \mathcal{D}} \left[ \underbrace{\log \pi(y|x)}_{\text{Likelihood}} + \underbrace{\lambda_N \cdot Q_\pi(x, y) - \log Z_\pi(x)}_{\text{Inference-Awareness}} \right], \tag{13}$$

Taking gradient of this objective function over $\theta \in \Theta$ implies

$\nabla_\theta \mathbb{E}_{(x,y) \sim \mathcal{D}} [\log \pi_\theta(y|x) + \lambda_N \cdot Q_{\pi_\theta}(x, y) - \log Z_{\pi_\theta}(x)]$

$= \mathbb{E}_{(x,y) \sim \mathcal{D}} [\nabla_\theta \log \pi_\theta(y|x)] + \lambda_N \cdot \nabla_\theta \mathbb{E}_{(x,y) \sim \mathcal{D}} [Q_{\pi_\theta}(x, y)] - \nabla_\theta \mathbb{E}_{x \sim \mathcal{D}} [\log Z_{\pi_\theta}(x)]$

$= \mathbb{E}_{(x,y) \sim \mathcal{D}} [\nabla_\theta \log \pi_\theta(y|x)] + \lambda_N \cdot \mathbb{E}_{(x,y) \sim \mathcal{D}} [\mathbb{E}_{y' \sim \pi_\theta} [\nabla_\theta \log \pi_\theta(y'|x) \cdot \sigma(r(x, y) - r(x, y'))]]$

$\quad - \mathbb{E}_{x \sim \mathcal{D}} [\nabla_\theta \log \mathbb{E}_{\pi_\theta(y|x)} [\exp(\lambda_N \cdot Q_{\pi_\theta}(x, y))]]$

$= \mathbb{E}_{(x,y) \sim \mathcal{D}} [\nabla_\theta \log \pi_\theta(y|x)] + \lambda_N \cdot \mathbb{E}_{(x,y) \sim \mathcal{D}} [\mathbb{E}_{y' \sim \pi_\theta} [\nabla_\theta \log \pi_\theta(y'|x) \cdot \sigma(r(x, y) - r(x, y'))]]$

$\quad - \mathbb{E}_{x \sim \mathcal{D}} \left[ \frac{\mathbb{E}_{\pi_\theta(y|x)} [\nabla_\theta \log \pi_\theta(y|x) \cdot \exp(\lambda_N \cdot Q_{\pi_\theta}(x, y))] + \mathbb{E}_{\pi_\theta(y|x)} [\nabla_\theta \exp(\lambda_N \cdot Q_{\pi_\theta}(x, y))]}{\mathbb{E}_{\pi_\theta(y|x)} [\exp(\lambda_N \cdot Q_{\pi_\theta}(x, y))]} \right].$

This further implies that

$\nabla_\theta \mathbb{E}_{(x,y) \sim \mathcal{D}} [\log \pi_\theta(y|x) + \lambda_N \cdot Q_{\pi_\theta}(x, y) - \log Z_{\pi_\theta}(x)]$

$= \mathbb{E}_{(x,y) \sim \mathcal{D}} [\nabla_\theta f(x, y; \theta)] - \mathbb{E}_{x \sim \mathcal{D}} \left[ \mathbb{E}_{\pi_\theta(y|x)} \left[ \frac{\exp(\lambda_N \cdot Q_{\pi_\theta}(x, y))}{\mathbb{E}_{\pi_\theta(y|x)} [\exp(\lambda_N \cdot Q_{\pi_\theta}(x, y))]} \cdot \nabla_\theta f(x, y; \theta) \right] \right],$

through collecting terms from the above expression and recalling the definition of $\nabla_\theta f(x, y; \theta)$ as

$\nabla_\theta f(x, y; \theta) := \nabla_\theta \log \pi_\theta(y|x) + \lambda_N \cdot \mathbb{E}_{y' \sim \pi_\theta} [\nabla_\theta \log \pi_\theta(y'|x) \cdot \sigma(r(x, y) - r(x, y'))].$

This further implies that

$$\nabla_\theta \mathbb{E}_{(x,y) \sim \mathcal{D}} [\log \pi_\theta(y|x) + \lambda_N \cdot Q_{\pi_\theta}(x, y) - \log Z_{\pi_\theta}(x)]$$
$$= \mathbb{E}_{(x,y) \sim \mathcal{D}} [\nabla_\theta f(x, y; \theta)] - \mathbb{E}_{x \sim \mathcal{D}, y \sim \pi_{bon}} [\nabla_\theta f(x, y; \theta)],$$

completing the proof of this lemma.

### A.3 PROOF OF LEMMA 2

Recall the RL objective function

$$\max_{\pi \in \Pi} J(\pi) := \mathbb{E}_{x \sim P, y \sim \pi_{\mathrm{bon}}(\cdot|x;\pi,r,N,T)}[R(x,y)]. \tag{14}$$

Applying the REINFORCE trick (Sutton et al., 2009b) to this problem over the BoN policy class and using the analgous argument from the proof of Lemma 1, we have the following expression for the policy gradient:

$$\mathbb{E}_{y \sim \pi_{\mathrm{bon}}(\cdot|x), x \sim \mathcal{D}}\left[\nabla_\theta \log \pi_{\mathrm{bon}}(y|x) \cdot R(x,y)\right]$$

$$=\mathbb{E}_{x \sim \mathcal{D}, y \sim \pi_{\mathrm{bon}}(\cdot|x)}\left[\nabla_\theta f\left(x,y;\theta\right) \cdot R(x,y)\right] - \mathbb{E}_{x \sim \mathcal{D}, y \sim \pi_{\mathrm{bon}}}\left[\nabla_\theta f\left(x,y;\theta\right)\right] \cdot \mathbb{E}_{y \sim \pi_{\mathrm{bon}}(\cdot|x), x \sim \mathcal{D}}\left[R(x,y)\right]$$

$$=\mathbb{E}_{x \sim \mathcal{D}, y \sim \pi_{\mathrm{bon}}(\cdot|x)}\left[\nabla_\theta f_\theta\left(x,y\right) \cdot (R(x,y) - b(x))\right]$$

$$=\mathbb{E}_{x \sim \mathcal{D}, y \sim \pi_{\mathrm{bon}}(\cdot|x)}\left[\nabla_\theta \log \pi_\theta\left(y|x\right) \cdot (R(x,y) - b(x))\right]$$

$$+ \lambda_N \cdot \mathbb{E}_{x \sim \mathcal{D}, y \sim \pi_{\mathrm{bon}}(\cdot|x), y' \sim \pi_\theta}\left[\nabla_\theta \log \pi_\theta\left(y'|x\right) \cdot \mathbf{1}\{r(x,y) \geqslant r(x,y')\} \cdot (R(x,y) - b(x))\right]$$

$$=\mathbb{E}_{x \sim \mathcal{D}, y \sim \pi_{\mathrm{bon}}(\cdot|x)}\left[\nabla_\theta \log \pi_\theta\left(y|x\right) \cdot (R(x,y) - b(x))\right]$$

$$- \lambda_N \cdot \mathbb{E}_{x \sim \mathcal{D}, y \sim \pi_{\mathrm{bon}}(\cdot|x), y' \sim \pi_\theta}\left[\nabla_\theta \log \pi_\theta\left(y'|x\right) \cdot \mathbf{1}\{r(x,y) < r(x,y')\} \cdot (R(x,y) - b(x))\right].$$

This completes the proof of this lemma.

### A.4 PROOF OF LEMMA 3

Using the log-likelihood trick, and plugging in the BoN distribution from Equation (8), the gradient of Equation (6) can be computed as

$$\mathbb{E}_{y \sim \pi_{\mathrm{bon}}(\cdot|x), x \sim \mathcal{D}}\left[\nabla_\theta \log \pi_{\mathrm{bon}}(y|x) \cdot R(x,y)\right] = \mathbb{E}_{y \sim \pi_{\mathrm{bon}}(\cdot|x), R(x,y)=1, x \sim \mathcal{D}}\left[\nabla_\theta \log \pi_{\mathrm{bon}}(y|x)\right]$$

$$=\mathbb{E}_{x \sim \mathcal{D}}\left[\mathbb{E}_{y \sim \pi_{\mathrm{bon}}(\cdot|x), R(x,y)=1,}\left[\nabla_\theta \log \pi_\theta(y|x)\right] + (1 - \mathbb{E}_{y' \sim \pi(\cdot|x)}\left[\mathbf{1}_{R(x,y')=0}\right]^N)\nabla_\theta \log \frac{1 - \mathbb{E}_{y' \sim \pi(\cdot|x)}\left[\mathbf{1}_{R(x,y')=0}\right]^N}{1 - \mathbb{E}_{y' \sim \pi(\cdot|x)}\left[\mathbf{1}_{R(x,y')=0}\right]}\right]$$

$$=\mathbb{E}_{x \sim \mathcal{D}}\left[\mathbb{E}_{y \sim \pi_{\mathrm{bon}}(\cdot|x), R(x,y)=1}\left[\nabla_\theta \log \pi_\theta(y|x)\right] - \frac{1 - \mathbb{E}_{y' \sim \pi(\cdot|x)}\left[\mathbf{1}_{R(x,y')=0}\right]^N}{1 - \mathbb{E}_{y' \sim \pi(\cdot|x)}\left[\mathbf{1}_{R(x,y')=0}\right]}\mathbb{E}_{y' \sim \pi}\left[\nabla_\theta \log \pi_\theta(y'|x) \cdot \mathbf{1}_{R(x,y')=1}\right]\right.$$

$$\left. + \mathbb{E}_{y' \sim \pi}\left[\nabla_\theta \log \pi_\theta(y'|x) \cdot \mathbf{1}_{R(x,y')=1}\right] \cdot N \cdot \mathbb{E}_{y' \sim \pi(\cdot|x)}\left[\mathbf{1}_{R(x,y')=0}\right]^{N-1}\right]$$

$$=\mathbb{E}_{x \sim \mathcal{D}}\left[\mathbb{E}_{y \sim \pi_{\mathrm{bon}}(\cdot|x), R(x,y)=1}\left[\nabla_\theta \log \pi_\theta(y|x)\right] \cdot \frac{N I_{\mathrm{ref}}(x)^{N-1}(1 - I_{\mathrm{ref}}(x))}{1 - I_{\mathrm{ref}}(x)^N}\right].$$

$$=\mathbb{E}_{x \sim \mathcal{D}}\left[\mathbb{E}_{y \sim \pi_{\mathrm{bon}}(\cdot|x), R(x,y)=1}\left[\nabla_\theta \log \pi_\theta(y|x)\right] \cdot \frac{N \cdot I_{\mathrm{ref}}(x)^{N-1}}{1 - I_{\mathrm{ref}}(x)^N}\right.$$

$$\left. - (-1) \cdot \mathbb{E}_{y \sim \pi_{\mathrm{bon}}(\cdot|x), R(x,y)=0}\left[\nabla_\theta \log \pi_\theta(y|x)\right] \cdot \frac{N \cdot I_{\mathrm{ref}}(x)}{1 - I_{\mathrm{ref}}(x)}\right]$$

### A.5 PROOF OF COROLLARY 4

Using the log-likelihood trick, and plugging in the BoN distribution from Equation (8), the gradient of problem Equation (6) can be computed as

$$\mathbb{E}_{x \sim \mathcal{D}}\left[\mathbb{E}_{y \sim \pi_{\mathrm{bon}}(\cdot|x), R(x,y)=1}\left[\nabla_\theta \log \pi_\theta(y|x)\right] \cdot \frac{N \cdot I_{\mathrm{ref}}(x)^{N-1}}{1 - I_{\mathrm{ref}}(x)^N}\right.$$

$$\left. + \mathbb{E}_{y \sim \pi_{\mathrm{bon}}(\cdot|x), R(x,y)=0}\left[\nabla_\theta \log \pi_\theta(y|x)\right] \cdot \frac{N \cdot I_{\mathrm{ref}}(x)}{1 - I_{\mathrm{ref}}(x)}\right]$$

$$=\mathbb{E}_{x \sim \mathcal{D}}\left[\mathbb{E}_{y \sim \pi_{\mathrm{bon}}(\cdot|x), R(x,y)=1}\left[\nabla_\theta \log \pi_\theta(y|x)\right] \cdot \frac{N I_{\mathrm{ref}}(x)^{N-1}(1 - I_{\mathrm{ref}}(x))}{1 - I_{\mathrm{ref}}(x)^N}\right].$$

---

**Algorithm 1** BoN-SFT

1: **Input:** Verifier score $r$, environment reward $R$, expert dataset $\mathcal{D}$
2: **for** $t = 1, 2, \ldots$ **do**
3:     Sample a batch of prompts and solutions $\{x_i, y_i\}_{i=1}^{B}$ from the expert data $\mathcal{D}$.
4:     **for** $i = 1, \ldots, B$ **do**
5:         Sample $N$ responses $\{y_{i,j}\}_{j=1}^{N}$ from $\pi_\theta(\cdot|x_i)$.
6:         Select the BoN response $y_i^* = \arg\max_j r(x_i, y_{i,j})$.
7:         Compute the gradient $\nabla_\theta f_\theta(x_i, y_i)$ using Equation (5).
8:     **end for**
9:     Update $\theta$ by following the gradient in Theorem 1 at learning rate $\alpha > 0$, i.e.,

$$\theta \leftarrow \theta + \alpha \left( \frac{1}{N} \sum_{i=1}^{N} [\nabla_\theta f(x_i, y_i; \theta)] - [\nabla_\theta f(x_i, y_i^*; \theta)] \right)$$

10: **end for**

---

## B PSEUDO-CODE AND IMPLEMENTATION DETAILS

Pseudo-code for all our SFT and RL methods is presented in Algorithms 1 to 4. Our implementation follows the standard use of an anchor policy, updated using exponential moving average. The policy is trained via BoN-aware losses, with additional KL divergence loss to the anchor policy. Table 2 shows the hyper-parameters used for all of our experiments.

We use linear annealing for the KL-coefficient. For all our RL experiments, we use a value baselines to reduce variance of our reward estimates. We normalize our advantage estimates w.r.t. the batch. For BoN-RLB the value network estimates $P_{\text{fail}}(x)$. We add additional clipping of the coefficients $g_N^+, g_N^-$ by clipping the value estimates for $P_{\text{fail}}$.

Table 2: Hyperparameters used in experiments.

| Hyperparameter | Value |
|---|---|
| Base model | Gemma 2b v2 |
| Optimizer | AdamW |
| Learning rate policy | 3e-6 |
| Policy warmup steps | 100 |
| Learning rate value | 1e-5 |
| Anchor EMA | 0.01 |
| Training steps | 2500 |
| Batch size | 32 |
| Sampling temperature | 1.0 |
| KL coefficient anneal steps | 2500 |
| KL coefficient anneal range | $1.0 \rightarrow 0.075$ |
| KL coefficient anneal delay | 10 |
| Clipping values for $P_{\text{fail}}$ | $\{0.01, 0.99\}$ |

### B.1 ANALYSIS OF BON-RLB WEIGHTS

The shifting balance between $g_N^+(p)$ and $g^-(p)$ with varying $p$ directly reflects the exploration-exploitation trade-off. As $p$ approaches 1, signifying very difficult problems, both $g_N^+(p)$ and $g^-(p)$ increase, but $g_N^+(p)$ rises more dramatically, especially for larger values of $N$. This sharp increase in $g_N^+(p)$ highlights the algorithm's increasing emphasis on learning from the few correct responses that are available in challenging scenarios. The effect is amplified by larger sample sizes: the more attempts are made, the more valuable the scarce successes become. The $\overline{g}_N^+(p)$ weight, used when only positive feedback is available, exhibits a similar upward trend with $p$ but with a less pronounced increase. This more moderate behavior can be attributed to the subtraction term in its formula, which tempers the influence of the positive samples and promotes a more balanced learning approach.

Finally, the potentially very large values of $g_N^+(p)$ for hard problems and larger $N$ introduce challenges for estimation. These high weights amplify the impact of individual positive samples, making

---

**Algorithm 2** BoN-RLB(P)

---
1: **Input:** Environment reward $R$, dataset $\mathcal{D}$
2: **for** $t = 1, 2, \ldots$ **do**
3:     Sample a batch of prompts $\{x_i\}_{i=1}^{B}$ from $\mathcal{D}$.
4:     **for** $i = 1, \ldots, B$ **do**
5:         Sample $N$ responses $\{y_{i,j}\}_{j=1}^{N}$ from $\pi_\theta(\cdot|x_i)$.
6:         Sample rewards for all candidate responses $\{R(x_i, y_i)\}_{i=1}^{N}$ from environment.
7:         Select the BoN response $y_i^* = \arg\max_j R(x_i, y_{i,j})$.
8:         Empirically estimate the base failure probability for each $x_i$, $i \in \{1, \ldots, B\}$,

$$\widehat{P}_{\text{fail}}(x_i) := \frac{1}{N} \sum_{j=1}^{N} \mathbf{1}_{R(x_i, y_{i,j})=0}.$$

9:     **end for**
10:    Update $\theta$ by following the gradient in Corollary 4 at learning rate $\alpha > 0$, i.e.,

$$\theta \leftarrow \theta + \alpha \left( \frac{1}{B} \sum_{i=1}^{B} \nabla_\theta \log \pi_\theta(y_i^{*,+}|x_i) \cdot \overline{g}^+(P_{\text{fail}}(x_i), N) \right)$$

    where $y_i^{*,+}$ represents the BoN sample that achieves a reward of 1.
11: **end for**

---

**Algorithm 3** BoN-RLB

---
1: **Input:** Environment reward $R$, dataset $\mathcal{D}$
2: **for** $t = 1, 2, \ldots$ **do**
3:     Sample a batch of prompts $\{x_i\}_{i=1}^{B}$ from $\mathcal{D}$.
4:     **for** $i = 1, \ldots, B$ **do**
5:         Sample $N$ responses $\{y_{i,j}\}_{j=1}^{N}$ from $\pi_\theta(\cdot|x_i)$.
6:         Sample rewards for all candidate responses $\{R(x_i, y_i)\}_{i=1}^{N}$ from environment.
7:         Select the BoN response $y_i^* = \arg\max_j R(x_i, y_{i,j})$.
8:         Empirically estimate the base failure probability for each $x_i$, $i \in \{1, \ldots, B\}$,

$$\widehat{P}_{\text{fail}}(x_i) := \frac{1}{N} \sum_{j=1}^{N} \mathbf{1}_{R(x_i, y_{i,j})=0}.$$

9:     **end for**
10:    Update $\theta$ by following the gradient in Lemma 3 at learning rate $\alpha > 0$, i.e.,

$$\theta \leftarrow \theta + \alpha \left( \frac{1}{B} \sum_{i=1}^{B} \nabla_\theta \log \pi_\theta(y_i^{*,+}|x_i) \cdot g^+(P_{\text{fail}}(x_i), N) - \nabla_\theta \log \pi_\theta(y^{*,-}|x_i) \cdot g^-(P_{\text{fail}}(x_i)) \right)$$

    where $y_i^{*,+}, y_i^{*,-}$ represent the BoN samples that achieve rewards of 1 and 0 respectively.
11: **end for**

---

the training more vulnerable to noise and potentially hindering convergence to a stable optimal policy. This underscores the need for techniques like gradient clipping or regularization to mitigate the destabilizing effects of high weight values and ensure robust learning, or alternatively, using $\overline{g}_N^+(p)$ with Corollary 4 to ensure boundness of the gradient weights.

---

**Algorithm 4** BoN-RL

---

1: **Input:** Verifier score $r$, environment reward $R$, dataset $\mathcal{D}$
2: **for** $t = 1, 2, \ldots$ **do**
3:      Sample a batch of prompts $\{x_i\}_{i=1}^{B}$ from $\mathcal{D}$.
4:      **for** $i = 1, \ldots, B$ **do**
5:          Sample $N$ responses $\{y_{i,j}\}_{j=1}^{N}$ from $\pi_\theta(\cdot|x_i)$.
6:          Select the BoN response $y_i^* = \arg\max_j r(x_i, y_{i,j})$.
7:          (If environment reward $R$ is available to the BoN algorithm, we replace verifier $r$ with that.)
8:          Sample the reward $R(x_i, y_i^*)$ from environment.
9:          Compute the gradient $\nabla_\theta f_\theta(x_i, y_i)$ using Equation (5).
10:      **end for**
11:      Update $\theta$ by following the gradient in Lemma 2 at learning rate $\alpha > 0$, i.e.,

$$\theta \leftarrow \theta + \alpha \left( \frac{1}{B} \sum_{i=1}^{B} \nabla_\theta f_\theta\left(x_i, y_i^*\right) \cdot \left(R(x_i, y_i^*) - b(x_i)\right) \right)$$

     where $b_\psi(x_i), i = 1, \ldots, B$ is a learned baseline value function of $\pi_{\text{bon}}$, i.e.,

$$\psi^* \in \arg\min_\psi \frac{1}{B} \sum_{i=1}^{B} [R(x_i, y_i^*) - b_\psi(x_i)]^2$$

12:      Update value estimate $\psi$ using the current environment reward target and BoN policy trajectories.
13: **end for**

---

## C    ALGORITHMIC EXTENSIONS

### C.1    ENTROPY-REGULARIZED RL

We would like to study an entropy-regularized RL problem for the $\pi_{\text{bon}}$ policy. Recall that generally in entropy-regularized RL, we solve

$$\max_{\pi(\cdot|x) \in \Delta} \mathbb{E}_{x \sim \mathcal{D}} \big[ \mathbb{E}_{y \sim \pi(\cdot|x)} [R(x, y)] - \beta \cdot KL(\pi || \pi_\beta)(x) \big], \tag{15}$$

where $R(x, y)$ is the environment reward (that is not necessarily identical to the verifier score model), $\pi_\beta$ is a baseline policy, and $\beta > 0$ is the weight for the KL regularization term. Using the consistency condition of KL-regularized MDP, solving for the optimal policy of this problem is equivalent to finding a solution pair of $V$ and $\pi \in \Delta$ of the following equation:

$$V(x) = R(x, y) + \beta \log \pi_\beta(y|x) - \beta \log \pi(y|x), \ \forall x \in \mathcal{D}, \ \forall y \tag{16}$$

Now, we further parameterize the policy variable $\pi$ with the BoN policy $\pi_{\text{bon}}$, then with the sufficiency part of the consistency condition one can show that $\pi_{\text{bon}}$ is an optimal RL policy of Equation (15) if there exists a pair of $V$ and $\pi$ that satisfies the following equation

$$V(x) = R(x, y) + \beta \log \pi_\beta(y|x) - \beta \left(\log \pi(y|x) + \lambda_N \cdot Q_\pi(y, x) - \log Z_\pi(x)\right), \ \forall x \in \mathcal{D}, \ \forall y \tag{17}$$

There are two ways to approximately find the solution in Equation (17). The first way is to reformulate the above equation with a condition that equates the values between any pairwise states and outputs $(x, y, y')$:

$$R(x, y') + \beta \log \frac{\pi_\beta(y'|x)}{\pi(y'|x)} + \beta \lambda_N \cdot Q_\pi(y', x) = R(x, y) + \beta \log \frac{\pi_\beta(y|x)}{\pi(y|x)} + \beta \lambda_N \cdot Q_\pi(y, x), \ \forall x \in \mathcal{D}, \ \forall y, y'. \tag{18}$$

Suppose one have access to pairwise labels in the data-set, then this formulation eliminates any terms that are independent to $y$ and circumvents the need of solving for the value function $V$. One may approximately solve Equation (18) by minimizing the following $\ell_2$ loss:

$$\min_{\pi \in \Delta} \mathbb{E}_{(x, y, y') \in \mathcal{D}} \left[ (g(x, y; \pi) - (g(x, y'; \pi))^2 \right],$$

$$g(x, y; \pi) := R(x, y) + \beta \log \frac{\pi_\beta(y|x)}{\pi(y|x)} + \beta \lambda_N \cdot Q_\pi(y, x).$$

This formulation is similar to that in IPO (Azar et al., 2024). However, unlike IPO, where the term $g(x, y; \pi)$ is linear in the logits of $\pi$ and therefore one can show that its $\ell_2$ minimization problem has a unique solution, in this case $g(x, y; \pi)$ also depends on $Q_\pi$, which is a function of $\pi$ (and

thus a nonlinear function of its logits), preventing us from drawing similar conclusions that the $\ell_2$ minimization problem has a unique solution. Therefore, even if one can exactly solve this $\ell_2$ minimization problem (and make the loss zero), there is no guarantee that the solution policy $\pi^*$ corresponds to the base policy of an optimal $\pi_{\text{bon}}$ policy to the KL-regularized RL problem.

For the second approach, consider the following linear programming reformulation of Equation (17):

$$\min_{V, \pi \in \Delta} \mathbb{E}_{x \in \mathcal{D}}[V(x)]$$

s.t. $V(x) \geqslant R(x, y) + \beta \log \pi_\beta(y|x) - \beta \left( \log \pi(y|x) + \lambda_N \cdot Q_\pi(y, x) - \log Z_\pi(x) \right), \ \forall x \in \mathcal{D}, \ \forall y$ (19)

Since the inequality constraint is a convex function in $\pi$ and an affine function in $V$, by strong duality it has the following equivalent Lagrangian-dual formulation:

$$\max_{\kappa(\cdot, \cdot) \geqslant 0} \min_{V, \pi \in \Delta} \mathbb{E}_{(x,y) \in \mathcal{D}} \left[ V(x) + \kappa(x, y) \cdot \left( R(x, y) + \beta \frac{\pi_\beta(y|x)}{\pi(y|x)} - \beta \left( \lambda_N \cdot Q_\pi(y, x) - \log Z_\pi(x) - V(x) \right) \right) \right]$$
$$= \max_{\kappa(\cdot, \cdot) \geqslant 0} \min_V \mathbb{E}_{\mathcal{D}} \left[ (1 - \kappa(x, y)) \cdot V(x) + \kappa(x, y) \cdot (R(x, y) + \beta \cdot \pi_\beta(y|x)) \right] - \max_{\pi \in \Delta} \mathbb{E}_{\mathcal{D}} \left[ \kappa(x, y) \cdot \log \pi_{\text{bon}}(y|x; \pi) \right]$$
(20)

This formulation can be viewed as an weighted-SFT approach that iteratively updates (i) the base policy $\pi$ that maximizes the likelihood of $\pi_{\text{bon}}$ over data $\mathcal{D}$, weighted with importance weights $\kappa(x, y)$, and (ii) the importance weight function $\kappa$ itself. Here, the value function $V(x)$ is simply an auxiliary variable.

## C.2 Improved Efficiency with BoN Distillation

While Lemma 1 provides a recipe for training a base policy to adapt to the BoN inference strategy, a key challenge lies in the computational cost and data inefficiency associated with BoN sampling, especially when $N$ is large. Particularly, each gradient update requires generating $N$ samples from the current base policy, which can be prohibitively expensive. Furthermore, using these samples solely for a single gradient update may deem wasteful.

To alleviate this issue, leveraging the recent advances in BoN Distillation (BoND) (Sessa et al., 2024), an RLHF algorithm that distills BoN behaviors into a standard LLM, we approximate the BoN distribution of the current $\pi$. This results in an iterative, two-step procedure. First, we estimate a BoND policy $\pi_{\text{BoND}}$ (parameterized by weights $\phi$) of $\pi$ by solving the distribution-matching problem: $\min_\phi \mathbb{E}_{x \sim \mathcal{D}}[\text{KL}(\pi_\phi || \pi_{\text{bon}})(x)]$, where the backward-KL metric induces quantile-based advantage and mode-seeking behaviors to $\pi_{\text{BoND}}$. Utilizing the variational form $\pi_{\text{bon}}(y|x) \propto \pi \cdot \exp(\lambda_N Q_\pi)(y|x)$, this problem can be further reformulated as

$$\pi_{\text{BoND}}(y|x) \in \arg\max_\phi \ \mathbb{E}_{x \sim \mathcal{D}}[\mathbb{E}_{y \sim \pi_\phi(\cdot|x)}[Q_\pi(y, x)] - \frac{1}{\lambda_N} \text{KL}(\pi_\phi || \pi)(x)]. \quad (21)$$

Second, equipped with the BoND policy, we change the gradient of Lemma 1 with the approximate gradient $\mathbb{E}_{(x,y) \sim \mathcal{D}}[\nabla_\theta f(x, y; \theta)] - \mathbb{E}_{x \sim \mathcal{D}, y \sim \pi_{\text{BoND}}(\cdot|x)}[\nabla_\theta f(x, y; \theta)]$. In general, this approach is also well-connected with Contrastive Divergence (Carreira-Perpinan & Hinton, 2005) in energy-based learning, which promotes the idea of approximately sample from the current target distribution ($\pi_{\text{bon}}$ in our case). It shows that the learning algorithm can still converge to an optimum w.r.t. the original objective function as long as the gradient estimated by the approximate samples still points at an ascending direction.

## C.3 Connection to STaR

Consider the popular STaR method (Zelikman et al., 2022) applied for training $\pi_{\text{bon}}$, which updates $\theta$ by following the reward-weighted gradient:

$$\mathbb{E}_{x \sim \mathcal{D}, y \sim \pi_{\text{bon}}(\cdot|x)} [\nabla_\theta \log \pi_\theta(y|x) \cdot R(x, y)]. \quad (22)$$

Notice that the policy gradient of BoN-RL is a sum of two terms: $\nabla_\theta J(\theta) = g_1(\theta) + g_2(\theta)$, where $g_1(\theta) = \mathbb{E}_{x \sim \mathcal{D}, y \sim \pi_{\text{bon}}(\cdot|x)} [\nabla_\theta \log \pi_\theta(y|x) \cdot R(x, y)]$ is equivalent to that of BoN-STaR, updating $\pi$ via weighted supervised fine-tuning over the responses and the rewards obtained by the current BoN policy, and $g_2(\theta) = \mathbb{E}_{x \sim \mathcal{D}, y \sim \pi_{\text{bon}}(\cdot|x)} [\nabla_\theta (\lambda_N Q_\pi - \log \mathbb{E}_\pi \exp(\lambda_N Q_\pi))(x, y) \cdot R(x, y)]$ accounts for the gradient effect of the importance sampling term $(\exp \lambda_N Q_\pi / Z_\pi)(x, y)$ between $\pi$ and $\pi_{\text{bon}}$, emphasizing on how much it can improve the reward. The additional $g_2(\theta)$ component

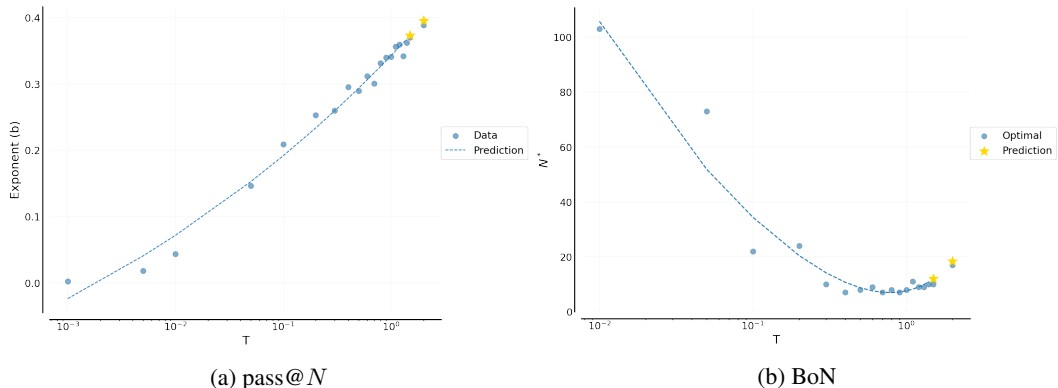

(a) pass@$N$          (b) BoN

Figure 9: Scaling of exponent w.r.t temperature in pass@$N$ and optimal $N$ w.r.t. temperature in BoN. Dashed curves denote in-training predictions, stars denote extrapolation values for the corresponding temperatures.

|  | pass@$N$ | BoN | MajorityVoting |
|---|---|---|---|
| Gemma-9B | 0.986 | 0.989 | 0.89 |
| Gemma-2B | 0.998 | 0.998 | 0.784 |

Table 3: R-squared values for different language models and inference algorithms.

makes BoN-RL amenable to the distributional shifts introduced by the BoN procedure, enabling the base policy to be adept at utilizing the BoN exploration mechanism to optimize the reward.

# D    EXPERIMENTAL DETAILS

## D.1    ADDITIONAL SCALING RESULTS WITH GEMMA 9B VERIFIER AND POLICY MODELS

Similar to Gemma2B co-scaling experiments, for Gemma 9B co-scaling, we present additional results in Figure 9. We analyze the optimal exponent $b^*(T)$ w.r.t different temperatures (see co-scaling in Section 5) and find that a power law functional form can explain the relationship very accurately, achieving very low extrapolation error for pass@$N$ and BoN, i.e., $2.75e-05$ and $2.87$, respectively. Results for pass@$N$ suggest that exponent can be accurately predicted from just temperature. We also inspect how optimal $N^*$ scales with $T$ in BoN. We fit a power law function plus a linear term which accurately predicts optimal $N$ for unseen temperatures. Predictions of the fitted model can be used to achieve close to optimal performance, achieving less than $0.001$ point drop in BoN performance. This suggests that our predictive model makes accurate predictions that keeps the optimal performance.

**Generalization of scaling predictions.** In Table 3, we compare various inference algorithms and LLMs of different sizes. For MajorityVoting algorithm, we use MC estimation to simulate different sample sizes. We use the same functional form used for co-scaling experiments in Section 5 for MajorityVoting. Our results remark the strong generalization performance of our predictive models across different LLMs (both policy and reward models) and inference algorithms. The same power-law function with a linear trend term applies well to both BoN and MajorityVoting. While we use unbiased estimates for pass@$N$ and BoN, we use MC estimation for MajorityVoting which leads to less smooth curves and slightly lower prediction performance.

**Gemma-9B Results.** In Figure 10, we present results for Gemma-9B policy and reward models. Using Gemma-9B improves both pass@$N$ and BoN significantly compared to Gemma-2B. We observe that the gap between using large temperatures (0.7 or 1.0) and very small temperatures (0.1) also increased. While Gemma-2B showed very strong reward model overoptimization for larger $N$ and temperatures, we see a lesser overoptimization for Gemma-9B models.

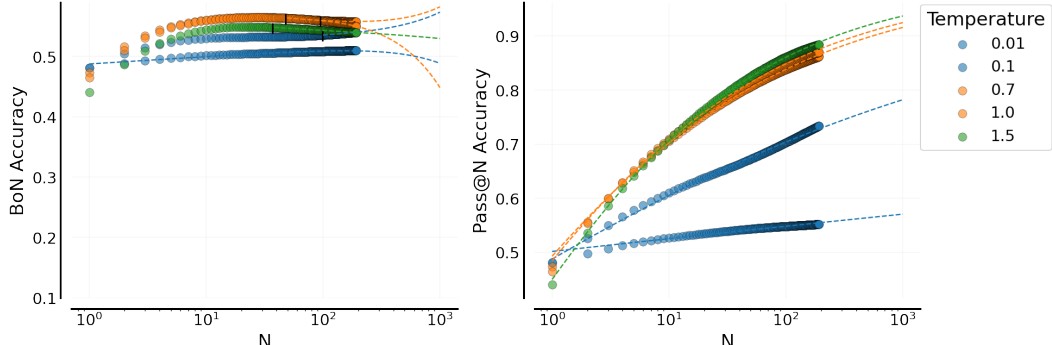

Figure 10: pass@$N$ (left) and BoN (right) performance for Gemma-9B. While curves show similar shape as Gemma-2B models, overall performance is globally improved and overoptimization is reduced.

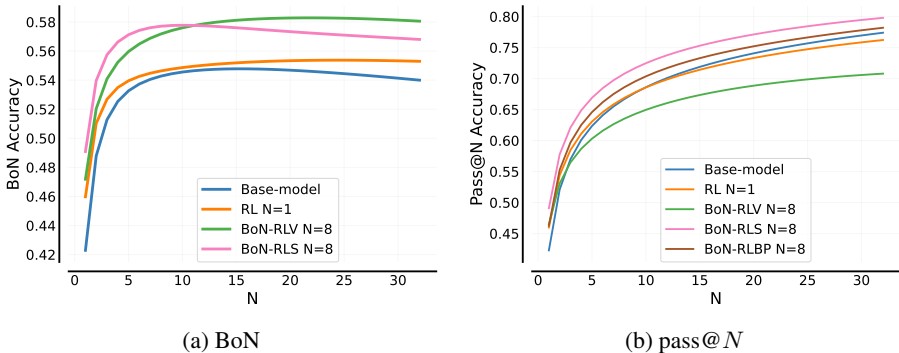

(a) BoN                                    (b) pass@$N$

Figure 11: BoN and pass@$N$ Accuracy Results on Hendrycks MATH with Gemma 9B.

### D.2 DETAILS OF BON-AWARE FINE-TUNING EXPERIMENTS

For the MATH benchmark, we trained the Gemma 2B and 9B models with the Hendrycks MATH dataset. Following Lightman et al. (2023), we augment the original 7500 MATH training problems with 4500 problems from the test set, evaluating performance on the remaining 500 problems. In the supervised setting, we leverage a larger Gemini 1.5 Flash model (Reid et al., 2024) to generate MATH solutions with answers and steps (32 candidates for each of the MATH problems), sub-sampling only the correct responses and distilling knowledge into the Gemma 2B model. In the RL setting, we use a binary environment reward denoting whether the model's answer matches the ground truth answer. The verifier used in all BoN experiments is a separate pre-trained Gemma 2B model that predicts the probability of a correct response given the prompt. The verifier is trained with the data collected from the Gemini 1.5 Flash model.

Alternatively, to benchmark our models on code generation, we train on MBPP (Austin et al., 2021) and evaluate on the HumanEval benchmark (Chen et al., 2021), following the standard procedures delineated in Kumar et al. (2024).

### D.3 ADDITIONAL BON-AWARE FINE-TUNING RESULTS

We now present additional results on BoN-aware fine-tuning (both SFT and RL).

### D.3.1 HENDRYCKS MATH WITH GEMMA 9B

We additionally benchmark a larger model, GemmaV2 9B, on Hendrycks MATH, with results shown in Figure 11. We observe that, similar to the trends of the experiments run with the Gemma 2B counterpart, BoN-RLV achieves the best BoN performance, while BoN-RL-S achieves the best pass@$N$ performance, with both substantially improving over the base model.

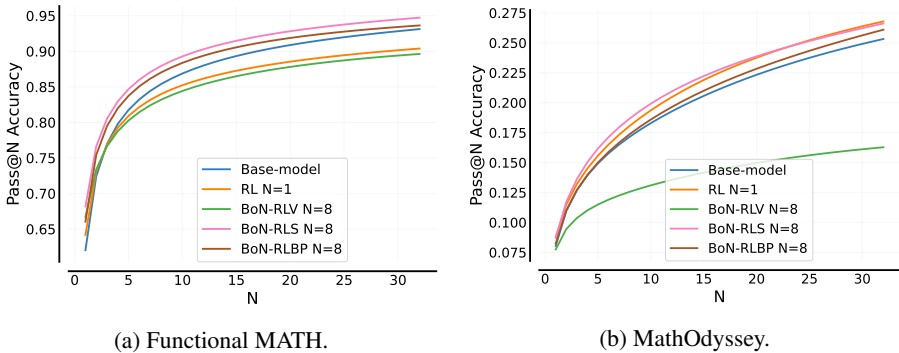

(a) Functional MATH.         (b) MathOdyssey.

Figure 12: pass@$N$ of BoN-RL and baselines on held-out benchmarks with Gemma 9B models.

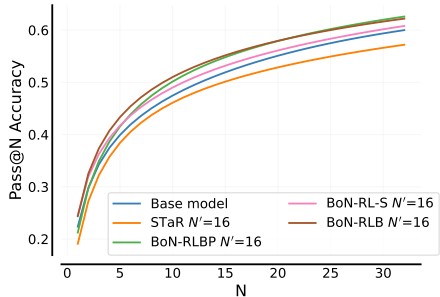

Figure 13: pass@$N$ of various methods with $N' = 16$

### D.3.2 HELD-OUT MATH BENCHMARKS WITH GEMMA 9B

To evaluate the generalization capabilities of our BoN-aware finetuned models, we additionally evaluate on two completely held-out and challenging benchmarks, Functional Math (Srivastava et al., 2024) and MathOdyssey (Fang et al., 2024). We present the results of these held-out benchmarks with 9B models in Figure 12, and observe that our fine-tuned models improve on both BoN and pass@$N$ for these held-out benchmarks similarly as in the case of Gemma 2B.

### D.3.3 PASS@N ACCURACY WITH DIFFERENT TRAINING SAMPLES, WITH GEMMA 2B

Similar to the experiments in Figure 6a, our BoN-RL-S, BoN-RLB, and BoN-RLB(P) models also demonstrate superior performance over the baseline methods in pass@$N$ evaluations with $N' = 16$ (Figure 13). In this case, STaR performs the worst (55% in pass@32) as it fails to (i) utilize negative samples in training for implicit exploration (unlike BoN-RLB, 60% pass@32), (ii) re-weight samples based on difficulty, prioritizing learning from challenging problems and avoiding overfitting to simpler ones (unlike BoN-RLB(P), 60% pass@32), and (iii) account for the importance sampling factor between the base policy and the BoN policy (unlike BoN-RL-S, 58% pass@32). BoN-RLB and BoN-RLB(P) are superior to BoN-RL-S on $N' = 16$ (Figure 13), but worse on $N' = 32$ (Figure 6a), suggesting that they suffer from instability with increasing $N'$. This is potentially due to the following observations: (i) The asymmetry between the positive ($g^+$) and negative ($g^-$) weights in BoN-RLB increases with $N'$, destabilizing its learning at larger $N'$ values; (ii) RL-BoN-S utilizes the variational approximation of $\pi_{\text{bon}}$ in its gradient update, introducing approximation errors that may cause its sub-optimal performance (relative to a stable instance of BoN-RLB trained at $N' = 16$); BoN-RLB(P) only uses positive samples, which inherits the shortcomings of STaR (lack of implicit exploration), yet it re-balances the examples with the difficulty of the problems. Overall, it leads to consistent yet mild performance degradation over BoN-RLB.

### D.3.4 COMPARING BON-RL WITH BASE-BON DISTILLATION BASELINES, WITH GEMMA 2B

We consider various alternative methods to improve Gemma 2B BoN accuracy through various data generation methods. We distill the Gemma 2B model using these datasets and compare to the

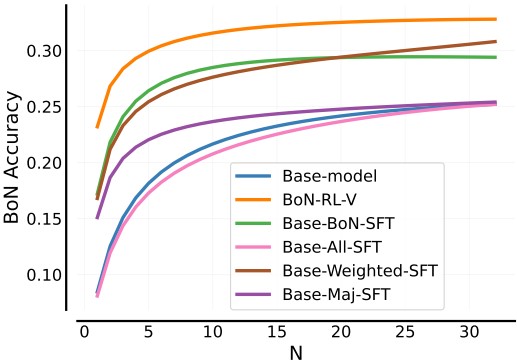

Figure 14: BoN Accuracy on MATH comparing Base Gemma 2B and BoN RL-V with other fine-tuning techniques: (1) BoN-SFT: Distillation of BoN sample for N=16; (2) All-SFT: Distillation of all N=16 samples (i.e., average sample); (3) Weighted-SFT: Distillation of all N=16 samples by average re-sampling w.r.t. verifier scores; and (4) Maj-SFT: Distillation of majority voting strategy.

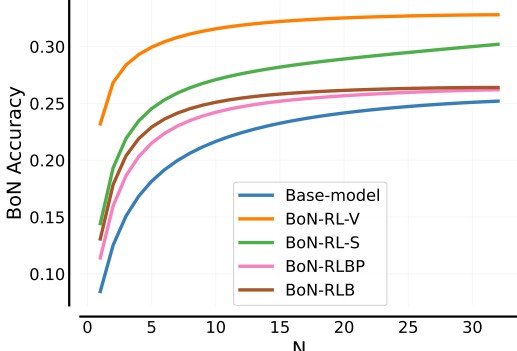

Figure 15: **Verifier Mismatch.** Plots show BoN accuracy under verifier-reward mismatch using Gemma 2B on MATH. During training verifier was used for BoN. On test, environment reward was used as verifier of the BoN strategy, inducing a mismatch in verifiers.

base Gemma 2B model and our BoN-RL-V method. We consider the following four distillation benchmarks (all run over Hendrycks MATH):

1. Base-BoN-SFT: In this method we generate a dataset of the best of $N = 16$ samples for each example in the dataset. We use the best sample as target to distill Gemma 2B.

2. Base-All-SFT: We use the full range of $N = 16$ samples as targets. This dataset is used to distill Gemma 2B to the average effective sample of the base model.

3. Base-Weighted-SFT: Similar to Base-All-SFT, we sample $N = 16$ samples for each example. We then re-sample $N = 16$ examples (from these samples, with repetition), weighted according to verifier scores. This dataset is used to distill Gemma 2B to the average effective sample, weighted by verifier scores.

4. Base-Maj-SFT: We use majority voting over $N = 16$ samples to select a target. We distill Gemma 2B to predict the majority voted target.

We show the BoN accuracy results of these methods in Figure 10. While the aforementioned baselines do improve BoN performance over the Base Gemma 2B model, they are still out-performed by our BoN-RL-V method, indicating the value of utilizing the inference BoN strategy explicitly during training.

