# OpenReview forum: "Inference-Aware Fine-Tuning for Best-of-N Sampling in Large Language Models"
_ICLR.cc/2025/Conference — ICLR 2025 Poster_

### Official Review · Reviewer_gx5H · 2024-10-27

**Soundness:** 2
**Presentation:** 3
**Contribution:** 3
**Rating:** 6
**Confidence:** 3

**Summary:**

This paper proposes an inference aware fine-tuning strategy for Best-of-N (BoN) to overcome the non-differentiable argmax operator for BoN. The proposed strategy incentivizes the model to balance the best and diverse responses for fine-tuning, achieves better performance compared to the base Gemma 2B model on the MATH dataset.

**Strengths:**

1. The incorporating of inference strategy into training is a compelling and well-founded idea.
2. The proposed BoN-aware fine-tuning to balance the trade-off between the best and diverse responses is well-motivated.

**Weaknesses:**

1. The experiments are conducted on a base model (Gemma 2B) and an evaluation dataset (MATH). More evaluation results on different base models and evaluation datasets are needed.
2. What is the underlying rationale for the BoN-SFT loss function presented in Equation (7)? The function appears to penalize y', even in scenarios where the sampled y' demonstrates high quality, as evidenced by r(x, y') approaching r(x, y). This raises a question: If the sampled y' exhibits desirable characteristics, wouldn't it be more beneficial for the policy model to learn from these high-quality responses?
3. It would be valuable to conduct a comparative analysis of BoN-SFT against several straightforward baseline approaches. These could include:

(1) Fine-tuning the base model using the highest-quality response selected from N generated samples.

(2) Fine-tuning the base model utilizing the ground truth responses.

(3) Fine-tuning the base model with all the sampled N responses.

(4) Fine-tuning the base model with the combination of all the sampled N responses and the ground truth responses.

**Questions:**

See "Weaknesses".

I would be happy to discuss further with the authors and reassess my score based on the rebuttal stage.

---

> ### Author Response · Authors · 2024-11-23
> **Individual Responses**
>
> Thank you for your comments. We address your points below.
>
> - Different Base Models and Datasets: In our revised paper, we added extensive experiments (i) using a larger 9B model (Figure 13), (ii) on another Fractional math and Math odyssey domains (Figure 14, 15, 16, 17), (iii) BoN-aware fine-tuning in the face of Verifier mismatch (Figure 12), (iv) on another HumanEval coding task (Figure 18), (v) other BoN distillation SFT baselines (Figure 11), as well as (vi) co-scaling studies with Gemma 9B model (Figure 9 and 10), for a more comprehensive analysis of our methods. See Appendix D3 in our updated paper and the results summarized in the above responses.
>
>
> - Rationale for Equation (7): The term penalizing y' in Equation (7) serves as a regularization factor. While learning from high-quality y' is important, preventing overfitting to the BoN sample is also crucial. This term balances the model's learning from the expert (y) and the BoN sample, ensuring robust generalization. It discourages the model from solely relying on maximizing the verifier score, as it should be “aware” of the way the model will be used in inference – by considering both the expert demonstration and the expected win-rate of the N generated solutions.
>
> - Comparative Analysis with Baselines:  We added new SFT experiments with baseline training datasets as suggested: (a) training on the best of N, (b) training on all N samples, and (c) training on N samples weighted by verifier scores, (d) labels selected with majority voting. See Figure 11 in Appendix D3 the updated paper and a summary of these results in the following.
>
> **BoN Accuracy**
> |  | Base-model | BoN-RL-V | Base-BoN-SFT | Base-All-SFT | Base-Weighted-SFT | Base-Maj-SFT |
> |---|---|---|---|---|---|---|
> | **N=1** | 18% | 24% | 21% | 10% | 18% | 15.5% |
> | **N=5** | 21% | 29% | 26.5% | 19% | 25.5% | 23% |
> | **N=10** | 22.5% | 31% | 28.5% | 22.5% | 27.5% | 25.5% |
> | **N=20** | 24% | 32% | 29.5% | 24.5% | 29% | 26.5% |
> | **N=30** | 24.5% | 32.5% | 30% | 25.5% | 29.5% | 27% |
>
> **Pass@N**
> |  | BoN-RL-S | BoN-RLB | Base-BoN-SFT | Base-All-SFT | Base-Weighted-SFT | Base-Maj-SFT |
> |---|---|---|---|---|---|---|
> | **N=1** | 16% | 17% | 18% | 08% | 18% | 16% |
> | **N=5** | 35% | 39% | 41% | 23% | 38% | 35% |
> | **N=10** | 45% | 49% | 50% | 32% | 46% | 42% |
> | **N=15** | 52% | 55% | 56% | 38% | 52% | 47% |
> | **N=20** | 57% | 60% | 59% | 43% | 56% | 51% |
> | **N=25** | 61% | 63% | 62% | 46% | 59% | 54% |
> | **N=30** | 64% | 65% | 64% | 48% | 61% | 56% |
>
>
> While the aforementioned baselines do improve BoN performance over the Base Gemma 2B model, they are still out-performed by our BoN-RL-V method, indicating the value of utilizing the inference BoN strategy explicitly during training. We believe these additions and clarifications significantly strengthen our paper and address your valuable feedback. We welcome further discussion during the rebuttal stage.

---

> > ### Comment · Reviewer_gx5H · 2024-11-25
> > **Response from Reviewer gx5H**
> >
> > I found that my concerns have been addressed. I have adjusted my rate accordingly.

---

> > > ### Author Response · Authors · 2024-11-25
> > > **Thanks**
> > >
> > > Thanks for your positive feedback and for the constructive comments that are pivotal to improve our work.

---

### Official Review · Reviewer_VBde · 2024-11-03

**Soundness:** 2
**Presentation:** 3
**Contribution:** 3
**Rating:** 6
**Confidence:** 2

**Summary:**

This paper presents a finetuning approach for large language models (LLMs) that incorporates the inference strategy directly into the model’s training process, with a particular focus on Best-of-N (BoN) inference. The authors argue that there is a disparity between the best-of-N policy typically used at inference time and the policy learned through standard supervised fine-tuning (SFT). To address this, they formalize an inference-aware fine-tuning problem that integrates the inference strategy into the training objective. Specifically, they develop a supervised BoN-aware fine-tuning method and a BoN-aware reinforcement learning (RL) with binary rewards, using policy gradients derived only from positive examples to mitigate data inefficiency. The proposed methods are evaluated on the Gemma 2B model, primarily using the Hendrycks MATH benchmark, with an emphasis on understanding the relationship between the number of samples and temperature.

**Strengths:**

- The paper addresses an important and timely problem, as best-of-N inference with verifiers has become increasingly popular for reasoning tasks.
- The paper provides a solid theoretical framework supporting the proposed methods with a clear formalization of inference-aware training objectives.

**Weaknesses:**

- The method is only evaluated on a single model and a single task, which limits insights into its broader applicability.
- Although BoN-aware fine-tuning is a key contribution, the experiments predominantly examine the relationship between the number of samples and temperature. This leans more toward an analysis of the proposed methods than a comparative assessment against strong baselines, leaving questions about its competitive advantages unanswered.

**Questions:**

1. Will inference-aware SFT and inference-aware RL reduce the model’s generalizability? How does the model perform when using beam search alone, without a verifier?
2. For inference-aware SFT experiments, what type of verifier is used? How does the method perform if there’s a mismatch between the training and testing verifiers?
3. In Figure 1, how is the empirical frequency calculated? What defines the best BoN performance? Also, what are the definitions of "easy" and "difficult" problems, given that the figure is based on the same set of MATH benchmark problems?

---

> ### Author Response · Authors · 2024-11-23
> **Individual Responses**
>
> Thank you for your positive feedback and insightful questions. We respond to your concerns as below.
>
> - Single Model and Task:  In our revised paper, we added extensive experiments (i) using a larger 9B model (Figure 13), (ii) on another Fractional math and Math Odyssey domains (Figure 14, 15, 16, 17), (iii) BoN-aware fine-tuning in the face of Verifier mismatch (Figure 12), (iv) on another HumanEval coding task (Figure 18), (v) other BoN distillation SFT baselines (Figure 11), as well as (vi) co-scaling studies with Gemma 9B model (Figure 9 and 10), for a more comprehensive analysis of our methods. See Appendix D3 in our updated paper and the results summarized in the above responses. All our experiments still showcase the superiority of our BoN inference-aware finetuning methods and their generalizability to broader problem settings.
>
> - Emphasis on N and T scaling over BoN-aware FT: Understanding the relationship between N and T is merely an initial crucial step to optimize BoN inference. Indeed, the main work and experiments of our paper develop BoN-aware fine-tuning of LLMs and demonstrate their effectiveness. To strengthen our comprehensive evaluation, on top of the existing Gemma 2B BoN-aware FT (SFT and RL) experiments and ablation studies on MATH, we also (i) added additional experiments on Gemma 9B model (Figure 9 and 10), (ii) tested our methods on held-out MATH benchmarks, e.g., Fractional math and Math Odyssey (Figure 14, 15, 16, 17), (iii) extended our work to solve HumanEval coding task (Figure 18), and provided additional BoN distillation SFT baselines (Figure 11) over the ones that we already included in the original paper (RLAIF with learned verifier scores, RLEF with system reward, STaR, SFT). This should further strengthen the evaluation of our BoN-aware fine-tuning methods and outline their competitive advantages.
>
>
> - Generalizability of Inference-Aware Methods: We have not observed a reduction in the model's generalizability with our inference-aware methods. In fact, as shown in Figure 4a, models trained with BoN-aware fine-tuning, e.g., BoN-RL-V that is trained with N'=32, can also improve performance at N=1 (pass@1), indicating improved generalizability not only on BoN policy but also on the base policy itself. Furthermore, as shown in Figure 14, 15, 16, 17 of the updated paper and the following results, our BoN-aware FT models that are trained on MATH manage to also perform well on other MATH domains, e.g., Fractional MATH. This also indicates the  generalizability of our BoN-are FT models on held-out benchmarks.
>
> **Gemma 2B Fractional Math: BoN Accuracy**
>
> |  | Base-model | RL N=1 | BoN-RLV N=16 | BoN-RLS N=16 |
> |---|---|---|---|---|
> | **N=1** | 14% | 36% | 44% | 26% |
> | **N=5** | 26% | 47% | 52% | 47% |
> | **N=10** | 31.5% | 51% | 55% | 52% |
> | **N=20** | 36% | 54% | 57% | 55% |
> | **N=30** | 38.5% | 55% | 58% | 56% |
>
> **Gemma 9B Fractional Math: BoN Accuracy**
> |  | Base-model | RL N=1 | BoN-RLV N=8 | BoN-RLS N=8 |
> |---|---|---|---|---|
> | **N=1** | 42.5% | 46% | 51% | 50% |
> | **N=5** | 51.5% | 55% | 57.5% | 57% |
> | **N=10** | 53% | 56% | 58% | 57.5% |
> | **N=20** | 53.5% | 56.5% | 58% | 58% |
> | **N=30** | 53.5% | 56.5% | 58% | 58% |
>
>
> - Verifiers and Mismatch: For our experiments, we used pre-trained Gemma 2B and 9B models as the verifier to predict pointwise correctness of responses (with 69% and 76% accuracy respectively). To understand how verifiers mismatch in training vs inference influence the performance of BoN policies, we added experiments comparing the test-time BoN performance of LLMs that were trained to align with the true underlying reward but were using a learned verifier in BoN inference. Details can be found in Figure 12 in the updated paper and a summary of numerical results is shown below, indicating the degree of performance degradation (over BoN-RL-V, a BoN-aware FT model that is both trained and tested with the same verifier).
>
> **Verifier reward mismatch experiments: BoN Accuracy**
>
> |  | Base-model | BoN-RL-V | BoN-RL-S | BoN-RLBP | BoN-RLB |
> |---|---|---|---|---|---|
> | **N=2** | 14% | 27% | 20% | 18% | 20% |
> | **N=5** | 21% | 30% | 25% | 23% | 24% |
> | **N=10** | 24% | 31.5% | 27.5% | 25% | 26% |
> | **N=20** | 25.5% | 32.5% | 29% | 26% | 26.5% |
> | **N=32** | 26% | 33% | 30% | 26.5% | 26.8% |
>
>
> - Figure 1 Calculation and Definitions: The empirical frequency in Figure 1 is calculated as the proportion of problems in the MATH 500 evaluation set for which a particular (N,T) pair achieved the highest accuracy. "Best BoN performance" is defined as the highest accuracy achieved by BoN on a given problem. "Easy" problems refer to those for which BoN achieves high accuracy with small T and N, indicating that extensive exploration is not necessary. Conversely, "difficult" problems require larger T for exploration and often larger N for effective exploitation. This distinction is based on the optimal (N,T) pairs found for each problem within the same MATH benchmark.

---

> ### Author Response · Authors · 2024-11-25
> **Reminder on author's responses**
>
> Hi Reviewer VBde,
>
> We have carefully considered your valuable feedback and have submitted a detailed response addressing the points raised in your reviews. We believe our response clarifies several aspects of the paper, highlights its contributions, and our additional work (attached above and in the updated paper) addresses your concerns. It would be great if you could take some time to review our responses and let us know your feedback.
>
> Thanks in advance,
> Authors of Paper 8453

---

> > ### Comment · Reviewer_VBde · 2024-11-28
> >
> > Thank the authors for their detailed reply. Most of my concerns are addressed and I will adjust my score accordingly.

---

> > > ### Author Response · Authors · 2024-12-01
> > > **Thank you**
> > >
> > > Thank you so much for providing your comments to improve our paper and for acknowledging our responses. Please feel free to let us know if you have further questions (if any), we are always happy to address that.

---

### Official Review · Reviewer_4VnU · 2024-11-04

**Soundness:** 2
**Presentation:** 2
**Contribution:** 2
**Rating:** 5
**Confidence:** 3

**Summary:**

The paper proposes a novel inference-aware fine-tuning paradigm, aiming to enhance the inference performance when scaling the compute.

To overcome the non-differential argmax operator within best-of-N (BoN), the paper proposes the first imitation learning and reinforcement learning methods for fine-tuning language models with BoN.

The experimental results show the BoN improvement of Gemma 2B on Hendrycks MATH from 26.8% to 30.8%, and Pass@K from 60% to 67%.

**Strengths:**

1. The studied problem is interesting and important: how to improve best-of-N sampling in the test time through advanced training methods applied to fine-tune the models.

2. The proposed method is reasonable and could be sound and useful if the method can be verified carefully and thoroughly.

**Weaknesses:**

- The paper writing can be improved: it is not easy to follow the core idea of the proposed methods. A visual Figure showing the core mechanism of the method is strongly suggested.

- The experiments are not sound enough to demonstrate the effectiveness of the proposed method, specifically:

**Regarding the co-scaling behavior of sample number N and temperature**: this analysis was conducted using only a single policy model and a single reward model.

How might the curves in Figure 2 appear if different language models or stronger reward models were used?

The conclusions drawn here cannot be readily generalized to other language models or reward models.


**The results presented in Figure 4 make it challenging to conclude that the proposed inference-aware fine-tuning significantly improves performance.**

One potential issue lies in the weakness of the reward model: the solution selected as having the highest reward is often inaccurate, and fine-tuning solely on this data may result in suboptimal model performance.

Selecting a broader range of solutions could benefit fine-tuning, as a correct answer might be found within these additional options.

Therefore, the observed improvement may stem from identifying a correct solution through an increased sample selection rather than from the inference-aware fine-tuning itself.


- I would also suggest conducting experiments on the alignment tasks and testing the models on AlpacaEval or Arena-Hard test sets.

**Questions:**

1. How is the accuracy of the used reward model?

2. How does best-of-N sampling relate to majority vote in math reasoning?

---

> ### Author Response · Authors · 2024-11-23
> **Individual responses 1**
>
> Thank you for your valuable feedback. We address your concerns as below.
>
> - Paper Writing and Visual Figure: We have revised the paper for improved clarity, focusing on a more intuitive explanation of our core ideas. We also added a visual schematic figure (see Figure 2) in the updated main paper and some discussions therein to improve the illustrations of our main idea of inference aware fine-tuning with BoN.
>
> - Co-scaling Behavior Analysis: In addition to the Gemma 2B co-scaling experiments, we also present results for Gemma-9B policy and reward models (see Figure 9 and 10 in the updated paper). Using Gemma-9B improves both Pass@N and BoN significantly compared to Gemma-2B. We observe that the gap between using large temperatures (0.7 or 1.0) and very small temperatures (0.1) also increased. While Gemma-2B showed very strong reward model over-optimization for larger N and temperatures, we see a lesser overoptimization for Gemma-9B models. Similar to Gemma2B co-scaling, for Gemma 9B co-scaling, we analyze the optimal exponent $b^*(T)$ w.r.t different temperatures for the functional form in Eq 5.1 and find that a power law functional form can explain the relationship very accurately, achieving very low extrapolation error for Pass@N and BoN (2.75e-05 and 2.87), suggesting that exponent can be accurately predicted from just temperature. We also inspect how optimal N* scales with T in BoN by fitting a power law function plus a linear term which accurately predicts optimal N for unseen temperatures. Predictions of the fitted model can be used to achieve close to optimal performance, achieving less than 0.001 point drop in BoN performance, suggesting that our predictive model makes accurate predictions that keeps the optimal performance.
>
> - Soundness of Experiments (Figure 4): We understand your concern about potential reward model weakness influencing the results. Our reward model, a separately pre-trained Gemma 2B (9B), has around 69% (76%) prediction accuracy on the MATH evaluation dataset. However, the significant improvement of BoN-RL-V over other baselines (e.g., BoN performance is over 30% for BoN-RL-V versus 22% for base Gemma 2B), including those using a broader range of solutions (e.g., RL-V), suggests that the gains are not solely due to increased sample selection. The inference-aware training itself plays a crucial role in enabling the model to better leverage BoN for performance improvement.
>
> - Extra tasks, e.g., AlpacaEval/Arena-Hard: We appreciate your suggestion for experiments on alignment tasks. While our current focus is on reasoning tasks (math and coding), exploring the applicability of our approach to alignment is an interesting future research direction. In our revised paper, we added extensive experiments (i) using a larger 9B model (Figure 13), (ii) on another Fractional math and Math odyssey domains (Figure 14, 15, 16, 17), (iii) BoN-aware fine-tuning in the face of Verifier mismatch (Figure 12), (iv) on another HumanEval coding task (Figure 18), (v) other BoN distillation SFT baselines (Figure 11), as well as (vi) co-scaling studies with Gemma 9B model (Figure 9 and 10), for a more comprehensive analysis of our methods. See Appendix D3 in our updated paper and the results summarized in our main response above.

---

> ### Author Response · Authors · 2024-11-23
> **Individual responses 2**
>
> - BoN and Majority Vote:  While both BoN and majority vote leverage multiple samples, BoN employs a learned verifier to select the best solution, while majority vote relies on the frequency of a particular output. To understand the similarities of these 2 inference methods, in Table 2 (Appendix D3) of the updated paper, we also added R-squared statistics of the performance betwen BoN and MajorityVoting algorithm, indicating strong correlation on performance across LLMs and these 2 inference algorithms.
>
> **R-square statistics of Gemma-2B/9B with Pass@N, BoN, and Majority Voting**
>
> | Model      | Pass@N | BoN Accuracy | MajorityVoting Accuracy |
> |------------|--------------------|-----------------|--------------------------|
> | Gemma-9B   | 98.6%              | 98.9%           | 89%                     |
> | Gemma-2B   | 99.8%              | 99.8%           | 78.4%                   |
>
> Furthermore, BoN, with its learned verifier, should have the better potential to capture more nuanced reasoning patterns compared to the simpler majority vote mechanism. Intuitively, suppose the base model would often produce diverse numerical answers, especially during the early stages of training when the probability of generating the correct answer is low. Then, majority voting may degenerate into random selection amongst mostly incorrect outputs, significantly limiting its ability to identify the correct solution. BoN, on the other hand, has the potential to capture higher quality solutions even with an imperfect verifier (absolute accuracy is less crucial to BoN, as long as the ordering of verifier scores can preserve ranking of the response quality). To support the above claim, our proposed BoN-aware RL fine-tuning manages to boost BoN and pass@N performance over models using majority voting, see Figure 11 in Appendix D3 and the following numerical results (Gemma 2B) for detailed comparisons.
>
> **BoN Accuracy**
> |  | BoN-RL-V | Base-Maj-SFT |
> |---|---|---|
> | **N=1** | 24% | 15.5% |
> | **N=5** | 29% | 23% |
> | **N=10** | 31% | 25.5% |
> | **N=20** | 32% | 26.5% |
> | **N=30** | 32.5% | 27% |
>
> **Pass@N**
> |  | BoN-RLB | Base-Maj-SFT |
> |---|---|---|
> | **N=1** | 17% | 16% |
> | **N=5** | 39% | 35% |
> | **N=10** | 49% | 42% |
> | **N=15** | 55% | 47% |
> | **N=20** | 60% | 51% |
> | **N=25** | 63% | 54% |
> | **N=30** | 65% | 56% |

---

> > ### Author Response · Authors · 2024-11-25
> > **Reminder on author's responses**
> >
> > Hi Reviewer 4VnU,
> >
> > We have carefully considered your valuable feedback and have submitted a detailed response addressing the points raised in your reviews. We believe our response clarifies several aspects of the paper, highlights its contributions, and our additional work (attached above and in the updated paper) addresses your concerns. It would be great if you could take some time to review our responses and let us know your feedback.
> >
> > Thanks in advance, Authors of Paper 8453

---

> > > ### Author Response · Authors · 2024-12-01
> > > **Official Comment by Authors**
> > >
> > > Hi Reviewer 4VnU,
> > > Thank you for your valuable inputs that help to improve our paper. We have incorporated your feedback, added additional experiments and explanations, and made improvements to the paper based on your suggestions. Since the authors' rebuttal period is ending soon, we kindly ask if you would consider revising the review score. Your further input would also be greatly appreciated.
> > >
> > > Cheers,
> > > Authors of Paper 8453

---

### Author Response · Authors · 2024-11-23
**General Response to All Reviewers 1**

We thank the reviewers for their insightful feedback and constructive criticism. We appreciate the reviewers acknowledging the importance and novelty of our work in addressing the gap between training and inference in large language models, particularly for Best-of-N (BoN) sampling. We have carefully considered all comments and made significant revisions to strengthen the paper. Specifically, we have:

1. Improved clarity and presentation by substantially revising the paper to enhance clarity and readability, as suggested by Reviewer 4VnU.  We have streamlined the presentation of our core ideas and provided additional explanations to make the paper more accessible. In the attached version of the paper we mark in red edits to the paper. We’ve also moved much mathematical notations to the appendix, to improve overall readability.

2. Expanded experiments by significantly expanding our experimental evaluation to address the concerns regarding limited model and task diversity. As requested, we have:
- Conducted further **co-scaling** experiments on **majority voting**, and with **9B Gemma model**. (Figure 9 and 10 in paper)
- Conducted **additional Distillation SFT experiments** using various baseline training datasets, as suggested by Reviewer gx5H. These include training on (a) the best of N samples, (b) all N samples as individual targets, and (c) N samples weighted by verifier scores. These experiments provide a more comprehensive analysis of our proposed BoN-SFT method.  (Figure 11 in paper)
- Added experiments illustrating how doing BoN-aware fine-tuning with the presence of **verifier mismatch** can impact BoN performance (Figure 12 in paper)
- Added results for the Hendrycks MATH tasks using a **9B parameter Gemma** model, showcasing the scalability of our approach to larger LLMs. (Figure 13 in paper)
- Added an additional **Fractional MATH and MATH Odyssey** benchmark for both **Gemma 2B and 9B** experiments. (Figure 14, 15, 16, 17 in paper)
- Included experiments on code generation, using the **MBPP and HumanEval coding datasets**, demonstrating the broader applicability of our method beyond mathematical reasoning. (Figure 18 in paper)

We summarize the highlighted results from the aforementioned new experiments below. Plots of all these all the new results are added to the updated paper in Appendix D3.

# Coding Benchmark
**Coding results trained on MBPP and tested on HumanEval**

| Metric   | Base model | RL-S N'=1 | BoN-RL-S N'=8 | BoN-RLBP N'=8 | BoN-RLB N'=8 |
|----------|----------------|----------------|--------------------|---------------------|-------------------|
| Pass@1  | 40.09%         | 39.37%         | 38.99%            | 41.12%              | 39.37%            |
| Pass@2  | 46.82%         | 45.94%         | 46.55%            | 48.34%              | 46.02%            |
| Pass@4  | 52.61%         | 51.15%         | 53.39%            | 54.98%              | 52.00%            |
| Pass@8  | 57.36%         | 55.57%         | 59.65%            | 61.09%              | 57.54%            |
| Pass@16 | 61.59%         | 59.76%         | 66.46%            | 67.07%              | 62.80%            |


# Gemma 9B on Hendrycks MATH

**BoN Accuracy**

|  | Base | SFT (N=1) | BoN-SFT N=8 | BoN-SFT N=32 |
|---|---|---|---|---|
| **N=1** | 42.5% | 44.5% | 49.5% | 43.1% |
| **N=5** | 51.5% | 53.5% | 55.3% | 55.5% |
| **N=10** | 53% | 54.5% | 55.8% | 56.3% |
| **N=20** | 53.5% | 54.5% | 55.8% | 56.2% |
| **N=30** | 53.5% | 54.3% | 55.7% | 56% |

|  | Base-model | RL N=1 | BoN-RLV N=8 | BoN-RLS N=8 |
|---|---|---|---|---|
| **N=1** | 42.5% | 46.5% | 47.5% | 49.5% |
| **N=5** | 51.5% | 54.5% | 57.5% | 56% |
| **N=10** | 53% | 55% | 57.8% | 57% |
| **N=20** | 53.5% | 55.3% | 58% | 57% |
| **N=30** | 53.5% | 55% | 58% | 56.8% |

**Pass@N Accuracy**

|  | Base | SFT (N=1) | BoN-SFT N=8 | BoN-SFT N=32 |
|---|---|---|---|---|
| **N=1** | 43% | 45.5% | 50% | 45.5% |
| **N=5** | 59% | 66% | 69.5% | 68.5% |
| **N=10** | 66% | 71% | 73.5% | 73% |
| **N=20** | 72% | 74% | 76% | 76% |
| **N=30** | 74.5% | 75.5% | 77% | 77.5% |

|  | Base-model | RL N=1 | BoN-RLV N=8 | BoN-RLS N=8 | BoN-RLBP N=8 |
|---|---|---|---|---|---|
| **N=1** | 43% | 46% | 48% | 49% | 47.5% |
| **N=5** | 59% | 63.5% | 66% | 68.5% | 67.5% |
| **N=10** | 66% | 70% | 71.5% | 74.5% | 73.5% |
| **N=20** | 72% | 75% | 75.5% | 78% | 77.5% |
| **N=30** | 74.5% | 77.5% | 78% | 79.5% | 79% |

---

> ### Author Response · Authors · 2024-11-23
> **General Response to All Reviewers 2**
>
> # Gemma 2B on Fractional MATH
>
> **BoN Accuracy**
>
> |  | Base | BoN-SFT N=8 | BoN-SFT N=4 | BoN-SFT N=16 | BoN-SFT N=32 |
> |---|---|---|---|---|---|
> | **N=1** | 14% | 15% | 13% | 14.5% | 15.5% |
> | **N=5** | 26% | 28.5% | 27% | 29% | 30% |
> | **N=10** | 31.5% | 34% | 32.5% | 34.5% | 35.5% |
> | **N=20** | 36% | 38.5% | 37% | 38.8% | 39.5% |
> | **N=30** | 38.5% | 40.5% | 39.5% | 40.8% | 41.5% |
>
> |  | Base-model | RL N=1 | BoN-RLV N=16 | BoN-RLS N=16 |
> |---|---|---|---|---|
> | **N=1** | 14% | 36% | 44% | 26% |
> | **N=5** | 26% | 47% | 52% | 47% |
> | **N=10** | 31.5% | 51% | 55% | 52% |
> | **N=20** | 36% | 54% | 57% | 55% |
> | **N=30** | 38.5% | 55% | 58% | 56% |
>
> **Pass@N Accuracy**
>
> | N | Base | BoN-SFT N=8 | BoN-SFT N=4 | BoN-SFT N=16 | BoN-SFT N=32 |
> |---|---|---|---|---|---|
> | **N=1** | 12% | 13% | 12% | 13.5% | 14% |
> | **N=5** | 25% | 28% | 26.5% | 29% | 30% |
> | **N=10** | 35% | 39% | 37% | 40% | 41% |
> | **N=20** | 44% | 48.5% | 46% | 49.5% | 50.5% |
> | **N=30** | 50% | 54% | 52% | 55% | 56% |
>
> |  | Base-model | BoN-RLV N=16 | BoN-RLS N=16 | S'TaR\_16 | BoN-RLBP N=16 | BoN-RLB N=16 |
> |---|---|---|---|---|---|---|
> | **N=1** | 12% | 40% | 22% | 29% | 21% | 27% |
> | **N=5** | 25% | 58% | 43% | 49% | 48% | 48% |
> | **N=10** | 35% | 66% | 56% | 60% | 61% | 60% |
> | **N=20** | 44% | 72% | 66% | 68% | 70% | 69% |
> | **N=30** | 50% | 75% | 71% | 72% | 73% | 73% |
>
> # Gemma 9B on Fractional MATH
>
> **BoN Accuracy**
>
> |  | Base-model | SFT N=1 | BoN-SFT N=8 | BoN-SFT N=32 |
> |---|---|---|---|---|
> | **N=1** | 42.5% | 44.5% | 49.5% | 43% |
> | **N=5** | 51.5% | 53.5% | 55.5% | 55.5% |
> | **N=10** | 53% | 54.5% | 56% | 56.5% |
> | **N=20** | 53.5% | 54.5% | 56% | 56% |
> | **N=30** | 53.5% | 54.5% | 55.5% | 56% |
>
> |  | Base-model | RL N=1 | BoN-RLV N=8 | BoN-RLS N=8 |
> |---|---|---|---|---|
> | **N=1** | 42.5% | 46% | 51% | 50% |
> | **N=5** | 51.5% | 55% | 57.5% | 57% |
> | **N=10** | 53% | 56% | 58% | 57.5% |
> | **N=20** | 53.5% | 56.5% | 58% | 58% |
> | **N=30** | 53.5% | 56.5% | 58% | 58% |
>
> **Pass@N Accuracy**
>
> |  | Base-model | SFT N=1 | BoN-SFT N=8 | BoN-SFT N=32 |
> |---|---|---|---|---|
> | **N=1** | 43.5% | 45.5% | 50% | 45.5% |
> | **N=5** | 62.5% | 66% | 69.5% | 68.5% |
> | **N=10** | 68.5% | 71% | 73.5% | 73% |
> | **N=20** | 72.5% | 74% | 76% | 76% |
> | **N=30** | 74.5% | 75.5% | 77% | 77.5% |
>
> |  | Base-model | RL N=1 | BoN-RLV N=8 | BoN-RLS N=8 | BoN-RLBP N=8 |
> |---|---|---|---|---|---|
> | **N=1** | 43.5% | 49% | 55% | 50% | 50% |
> | **N=5** | 62.5% | 68% | 72% | 70% | 70% |
> | **N=10** | 68.5% | 73% | 76% | 75% | 75% |
> | **N=20** | 72.5% | 77% | 79% | 78% | 78% |
> | **N=30** | 74.5% | 78% | 80% | 79% | 79% |
>
> # SFT Baselines Gemma 2B (requested by reviewer gx5H)**
> 1. BoN-SFT runs fine-tuning on best-of-N sample (N=16)
> 2. All-SFT runs fine-tuning on all N samples (N=16)
> 3. Weighted-SFT runs fine-tuning on a verifier weighted version of all N samples (N=16)
> 4. Maj-SFT runs fine-tuning on majority voting target of samples
>
> **BoN Accuracy**
> |  | Base-model | BoN-RL-V | Base-BoN-SFT | Base-All-SFT | Base-Weighted-SFT | Base-Maj-SFT |
> |---|---|---|---|---|---|---|
> | **N=1** | 18% | 24% | 21% | 10% | 18% | 15.5% |
> | **N=5** | 21% | 29% | 26.5% | 19% | 25.5% | 23% |
> | **N=10** | 22.5% | 31% | 28.5% | 22.5% | 27.5% | 25.5% |
> | **N=20** | 24% | 32% | 29.5% | 24.5% | 29% | 26.5% |
> | **N=30** | 24.5% | 32.5% | 30% | 25.5% | 29.5% | 27% |
>
> **Pass@N**
> |  | BoN-RL-S | BoN-RLB | Base-BoN-SFT | Base-All-SFT | Base-Weighted-SFT | Base-Maj-SFT |
> |---|---|---|---|---|---|---|
> | **N=1** | 16% | 17% | 18% | 08% | 18% | 16% |
> | **N=5** | 35% | 39% | 41% | 23% | 38% | 35% |
> | **N=10** | 45% | 49% | 50% | 32% | 46% | 42% |
> | **N=15** | 52% | 55% | 56% | 38% | 52% | 47% |
> | **N=20** | 57% | 60% | 59% | 43% | 56% | 51% |
> | **N=25** | 61% | 63% | 62% | 46% | 59% | 54% |
> | **N=30** | 64% | 65% | 64% | 48% | 61% | 56% |
>
>
> # Verifier reward mismatch experiments (requested by reviewer VBde)**
>
> **BoN Accuracy**
>
> |  | Base-model | BoN-RL-V | BoN-RL-S | BoN-RLBP | BoN-RLB |
> |---|---|---|---|---|---|
> | **N=2** | 14% | 27% | 20% | 18% | 20% |
> | **N=5** | 21% | 30% | 25% | 23% | 24% |
> | **N=10** | 24% | 31.5% | 27.5% | 25% | 26% |
> | **N=20** | 25.5% | 32.5% | 29% | 26% | 26.5% |
> | **N=32** | 26% | 33% | 30% | 26.5% | 26.8% |
>
> # R-square statistics of Gemma-2B/9B with Pass@N, BoN, and Majority Voting (requested by reviewer 4VnU)**
>
> | Model      | Pass@N | BoN Accuracy | MajorityVoting Accuracy |
> |------------|--------------------|-----------------|--------------------------|
> | Gemma-9B   | 98.6%              | 98.9%           | 89%                     |
> | Gemma-2B   | 99.8%              | 99.8%           | 78.4%                   |
>
>
> We believe these additions significantly strengthen the paper and address the key concerns raised by the reviewers. We detail our specific responses to each reviewer below.

---

### Meta-Review · Area_Chair_LLgH · 2024-12-19

**Metareview:**

This paper presents an approach for finetuning LLMs such that they are inference-strategy aware, and find that policies resulting from this strategy are more amenable to inference-time scaling.

On the plus side, this paper presents a rigorous formulation of an important problem (and quite timely, given the focus on inference-time compute these days) along with a sensible relaxation of the objective that enables tractable learning. On the negative side, experiments are only conducted on basic mathematical reasoning benchmarks, which may not generalize to more realistic settings.

**Additional Comments On Reviewer Discussion:**

Many reviewers pointed out that the original submission was very limited in scope, in particular focusing on a single model/benchmark. During the rebuttal phase, the authors conducted experiments across more models and benchmarks, which resulted in several reviewers changing their scores favorably. These results make the paper substantially more robust, which pushes this above the acceptance bar in my opinion.

---

### Decision · Program_Chairs · 2025-01-22

Accept (Poster)